# Confirmation bias through selective readout of information encoded in human parietal cortex

Hame Park[1,7] ✉, Ayelet Arazi[1,7], Bharath Chandra Talluri [1,2,7], Marco Celotto[3,4], Stefano Panzeri [3,8], Alan A. Stocker [5,8] & Tobias H. Donner [1,6,8] ✉

Decision-makers often process new evidence selectively, depending on their current beliefs about the world. We asked whether such confirmation biases result from biases in the encoding of sensory evidence in the brain, or alternatively in the utilization of encoded evidence for behavior. Human participants estimated the source of a sequence of visual-spatial evidence samples while we measured cortical population activity with magnetoencephalography. Halfway through the sequence, participants were prompted to judge the more likely source category. We find that processing of subsequent evidence depends on its consistency with the previously chosen category. Evidence encoded in parietal cortex contributes more to the estimation report when that evidence is consistent with the previous choice compared to when it contradicts that choice. Our results indicate that information contradicting pre-existing beliefs has little impact on subsequent behavior, despite being precisely encoded in the brain. This provides room for deliberative control to counteract confirmation biases.

Adaptive behavior in an uncertain world often requires agents to continuously update their beliefs about the state of the environment based on sequences of noisy observations ("evidence")[1–5]. A prominent bias shaping this process is confirmation bias, the tendency to selectively gather and interpret new evidence that supports a pre-existing belief[6,7]. Confirmation bias affects important areas of human judgment and decision-making including medical diagnostics, judicial reasoning, or scientific hypothesis testing[7]. Remarkably, confirmation bias also manifests in elementary, value-neutral decisions under uncertainty, such as judging the direction of a weak visual motion signal embedded in noise: once participants commit to a categorical judgment about the source of a noisy evidence stream, subsequent evidence that is consistent with that initial judgment tends to have a stronger impact on a subsequent decision compared to inconsistent evidence[8].

Recent advances in behavioral psychophysics have opened the door for a quantitative, within-participant assessment of the mechanisms giving rise to confirmation biases[8,9]. In principle, two distinct operations could give rise to selective processing of sensory evidence observable at the level of behavior. First, the encoding of sensory evidence could be modulated such that its precision depends on the consistency of the evidence with a previous decision or belief. Second, the downstream "readout" of that sensory representation for a

[1]Section Computational Cognitive Neuroscience, Department of Neurophysiology and Pathophysiology, University Medical Center Hamburg-Eppendorf, Martinistraße 52, Hamburg 20251, Germany. [2]Laboratory of Sensorimotor Research, National Eye Institute, National Institutes of Health, Bethesda, USA. [3]Institute for Neural Information Processing, Center for Molecular Neurobiology, University Medical Center Hamburg-Eppendorf, 20251 Hamburg, Germany. [4]Department of Brain and Cognitive Sciences, Picower Institute for Learning and Memory, Massachusetts Institute of Technology, Cambridge, MA 02139, USA. [5]Department of Psychology, University of Pennsylvania, 3710 Hamilton walk, Philadelphia, PA 19106, USA. [6]Bernstein Center for Computational Neuroscience Berlin, Humboldt-University Berlin, Philippstr. 13, Haus 6, 10115 Berlin, Germany. [7]These authors contributed equally: Hame Park, Ayelet Arazi, Bharath Chandra Talluri. [8]These authors jointly supervised this work: Stefano Panzeri, Alan A Stocker, Tobias H. Donner. ✉e-mail: mail.hamepark@gmail.com; t.donner@uke.de

subsequent behavioral decision may be altered, such that belief-inconsistent evidence contributes less to the decision than belief-consistent evidence. In either case, evidence consistent with a previous decision or belief would have stronger impact on later decisions than inconsistent evidence, but through different mechanisms. Previous work did not distinguish between these mechanisms. This distinction has important implications: a reduction in the precision of the encoding of inconsistent evidence implies unrecoverable information loss, while a consistency-dependent modulation of evidence readout is more malleable to strategic control. The latter mechanism could be counteracted, on the fly (within a given decision) or across longer timescales during learning.

The goal of the present study was to arbitrate between a selective encoding and a selective read-out mechanism of confirmation bias. To this end, we developed an approach for interrogating the encoding of sensory evidence in human cortical population activity, as well as the readout of this neural evidence code for behavior. Our approach was motivated by models of perceptual decision-making[10,11], according to which sensory evidence is first encoded in posterior areas of the cortical sensory-motor axis and then read out through a process including temporal accumulation in downstream brain areas[4,12–15]. Recent advances in neural recording and analysis now enable a dissection of these neural operations through the analysis of fine-grained patterns of human cortical population activity through magnetoencephalography (MEG) source imaging[16] and statistical techniques for quantifying evidence encoding and evidence readout[17,18]. Here, we combined such an MEG assessment of cortical population codes with a behavioral

task tailored to the precise quantification of selective processing of a protracted evidence stream. The task afforded a temporally precise quantification of the impact of evidence samples occurring at different lags from an intermediate categorical choice, on a final decision based on the complete evidence stream. Our findings support the selective readout scenario and refute the selective encoding scenario.

## Results

During MEG recordings, 30 healthy participants performed a continuous visual decision-making task (Fig. 1a). On each trial, they monitored a sequence of twelve evidence samples (checkered discs varying only in angular position) and finally reported their continuous estimate of the evidence source (i.e., mean of generative distribution, Fig. 1b). In the main experimental condition (dubbed "Choice" condition), participants were prompted to judge the category of the evidence source (left versus right from a reference line) after the sixth sample (Fig. 1a; mean accuracy: 81.4%, Fig. 1d, black). Sorting the subsequent evidence samples by their consistency with the previous choice enabled us to identify its impact on the processing of subsequent evidence samples, at behavioral and neural levels.

Previous psychophysics work reported choice-induced biasing effects on subsequent evidence processing that were reminiscent of the known effects of selective attention[8], and biases in continuous decisions induced by categorical choices as well as visual cues[19]. We, therefore, included a control condition ("Cue"), without an intermittent choice, but instead a cue after six samples, which informed participants about the true source category (left versus right from reference) with 75%

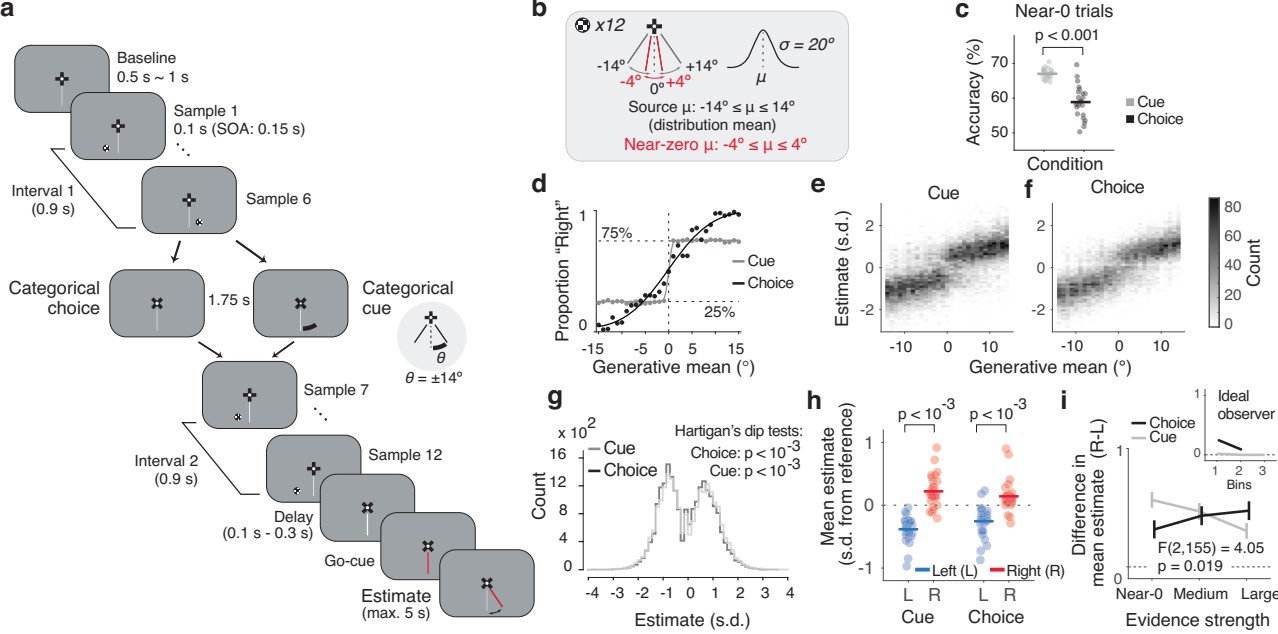

**Fig. 1 | Task design and behavior. a** Schematic sequence of events in example trial. Visual evidence samples (polar angles from vertical reference line, of small discs at 5° of eccentricity) are drawn from Gaussian distributions. After the sequence, participants report their inference about the underlying source (distribution mean) by moving a joystick. After six samples, participants either report a categorical judgment about the source, or receive a categorical cue (see main text). After cue presentation, participants make a pre-assigned (left or right, blocked) button press, to match Choice and Cue conditions in terms of motor output. **b** Range of generative means (μ) and standard deviation (σ). Red font indicates the near-zero generative mean trials used for consistency effect analyses (−4° to 4°) **(c)** Accuracy of the intermediate choice and reliability of the cue for the near-zero trials indicated in (**b**). **d** Proportion of "right" choices as a function of generative mean (Choice; first 6 samples). For comparison, the fraction of "right" Cues are plotted against the first 6 sample mean for Cue. Lines: fits of a sigmoid function on group

data. Dots are group averages. **e–i** Quantification of estimation biases. **e** Distributions of estimates as a function of generative mean for Cue condition. Estimates were first z-scored and binned before pooling probability densities across participants. **f** Same as (**e**), for Choice. **g** Distributions of estimates collapsed across generative means for Cue (gray) and Choice (black). P-values, Hartigan's dip test of unimodality. **h** Mean estimates for trials with 'left' and 'right' choice or cue categories on near-zero trials. Data points, participants; p-values, two-sided permutation tests. **i** Difference of mean estimates between 'right' and 'left' categories (as in (**h**)), as function of the strength of category evidence (binned). Lines, group average; error bars, SEM; statistics: interaction condition x evidence strength, two-factorial ANOVA, $\eta p^2 = 0.05$; 95% CI: [−0.096, 0.195]). Inset, ideal observer prediction (see main text). $N = 30$ participants for all panels except (**c, e, i**): $N = 27$ (excluding three participants with different spacing of generative means not suited for the corresponding analysis; Methods).

reliability (Fig. 1a). Prompting an intermittent choice or presenting an intermittent cue of similar reliability allowed us to determine if any biases induced by these two categorical events would depend on whether the event was self-generated (Choice) or externally provided (Cue).

Participants' categorical choices and continuous estimation reports tracked the source (i.e., generative mean) in a lawful manner (Fig. 1d–f; mean squared error of estimation reports: 5.93°). Even so, the estimation reports were also systematically biased, evident as different modes in the estimation distributions skewed away from the reference in both directions (Fig. 1e–g and Supplementary Fig. 1). We found significant deviation from unimodality for the group-level estimation distributions from Choice and Cue conditions (p-values in Fig. 1g) as well as in each individual for Choice, and all but one participant for Cue (Supplementary Fig. 1).

Bias was also evident when conditioning the mean estimates on the category of the previous choice or cue (Fig. 1h, i and Supplementary Fig. 2). For the Choice context, conditioning on the selected category results in some consistency bias due to the correlation between (sampling and internal) noise and the choice (see ideal observer prediction in Fig. 1i, inset). However, this does not apply for the Cue context, since the cue was coupled to the source (i.e., generative mean, Fig. 1i, inset). The difference in mean estimates between right and left categories of both conditions indicates that the intermittent categorical events (choice or cue) biased the final estimates to be consistent with that category. This matches previous results for retrospective estimation reports (see[19] and Discussion). Further, in the Choice context, the magnitude of this category-dependent difference in mean estimates tended to increase with the strength of the category evidence and conversely for Cue (Fig. 1i; Supplementary Fig. 2d). This pattern deviates from the pattern produced by a noise-free and unbiased observer (see inset and Methods) and is consistent with the notion that confirmation bias scales with confidence[9]: the accuracy of the choice (and the associated confidence[20]) increased with the strength of category evidence, while participants knew that the accuracy of the cue was independent of evidence strength (Fig. 1d).

In sum, participants' final estimation reports were biased, with the categorical events halfway through the trial being one factor inducing these biases. We next devised a temporally-specific approach to isolate the impact of these categorical events on the subsequent weighting of evidence that is postulated (for the interim choice) by psychological accounts of confirmation bias[7].

## Selective weighting of sensory evidence in a continuous decision

We quantified the weighting of the sequentially presented evidence in the final decision in terms of the mutual information between the stimulus samples (S), for each sample position in the sequence, and the behavioral estimation report (E). This is an information-theoretic quantification of the so-called "psychophysical kernel"[4,13,21] (Supplementary Fig. 3), which we denote as I(S;E). We used mutual information to comprehensively quantify the strength of the weight of evidence samples on behavioral estimates, capturing both linear and non-linear associations (see Supplementary Fig. 4 for a linear regression version of the analysis). To minimize the impact of the systematic trial-by-trial variations of generative means on I(S;E) and to isolate the impact of stochastic fluctuations in the evidence on the final estimation decision[21], we only included trials with generative means around zero (Fig. 1b). Simulations of an ideal (non-biased and noise-free) observer model confirmed that this procedure eliminated possible spurious effects originating from the correlations between the generative mean, cue or choice, and the final estimation, while ensuring enough data for computation of the information metrics (Methods). The same subset of trials was used in all comparisons of information measures between consistent and inconsistent samples reported below.

I(S;E) was significantly larger than zero for each sample, in both Choice and Cue conditions (Supplementary Fig. 3a, b), in line with the notion that participants integrated information across the sample sequence in their final continuous decisions, as is commonly observed for categorical decisions[4,13,21]. There was no difference in I(S;E) between samples in the first interval that were consistent versus inconsistent with the subsequent choice or cue (Fig. 2a, b, samples 1-6). Critically, however, in the second interval, I(S;E) was larger for consistent than for inconsistent samples (Fig. 2a, b: black bars) for samples immediately following the cue/choice. This indicates consistency-dependent, selective processing of sensory evidence induced by the preceding categorical choice or cue. The consistency effect on evidence weighting was stronger in the Choice compared to the Cue condition (Fig. 2c). Within the subset of trials close to the category boundary analyzed here, the fraction of correct choices was slightly smaller than the fraction of correct (i.e., valid) cues (Cue: 67.1% vs. Choice: 58.9%), when correctness was based on the generative mean (Fig. 1c, Wilcoxon sign-rank test). When computing psychophysical kernels using linear regression instead of I(S;E), we also found stronger weights on the final estimate for consistent compared to inconsistent samples following the cue/choice, and again a stronger consistency effect in the Choice than the Cue condition (Supplementary Fig. 4).

These behavioral results are in line with recent psychophysics work on choice consistency-dependent evidence re-weighting in other tasks[8,19]. The results extend previous work by delineating the precise time-course (across evidence samples) of the consistency effect and establishing such a consistency effect, albeit weaker, also for external

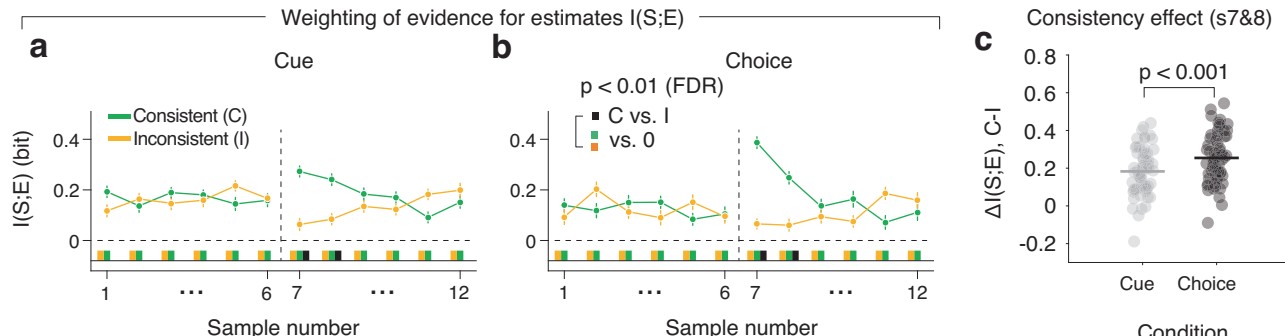

**Fig. 2 | Weight of perceptual evidence samples on choice. a, b** Impact of "sensory evidence" (samples) on estimation reports, computed as I(S;E), the mutual information between sample (S) and estimation report (E), for Cue (**a**) and Choice (**b**) conditions, for the entire stimulus sequence. I(S;E) is evaluated dependent on the consistency of each sample with the preceding (upcoming) choice or cue in the 2nd interval (1st interval). Data points; group average. error bars, SEM. Colored and black bars below 0, significant differences from 0 or between consistent and inconsistent, respectively (two-sided permutation tests, p < 0.01, FDR corrected). **c** Comparison of consistency effect between Choice and Cue (from (**a**, **b**)), for samples 7 and 8. Data points, participants; p-value, two-sided permutation test. N = 30 participants for all panels.

categorical cues. A model of neural populations at the encoding stage (Supplementary Fig. 5), identified three basic mechanisms that could produce this behavioral effect of consistency (Supplementary Fig. 5d, e: see smaller bar graphs of I(S;E) for inconsistent compared to consistent samples in first column): a reduction in the encoding precision for inconsistent samples (Supplementary Fig. 5d), or a reduction in the readout of the information about inconsistent samples encoded properly in neural activity (Supplementary Fig. 5e, f). More specifically, reduced readout of the encoded information could be due to an increase in noise in the decoding process (Supplementary Fig. 5e) or due to a mismatch between the neural activity patterns encoding the stimulus and the weights used to read out this stimulus representation for behavior (Supplementary Fig. 5f).

Critically, all three mechanisms gave rise to identical consistency effects at the level of behavior, that is larger weights for consistent than inconsistent samples in the psychophysical kernels (Supplementary Fig. 5d–f: first column). But at the level of neural activity, they produce different measures of neural encoding versus neural readout (Supplementary Fig. 5d–f, second to rightmost column). To distinguish between these alternative neural mechanisms that may underlie the observed consistency effects on evidence weighting, we, therefore, went on to dissect the encoding and readout of individual evidence samples in the brain.

### Encoding and readout of sensory evidence across cortex

We again used information theoretic measures to comprehensively quantify the neural encoding and readout of sample stimulus information in cortical population activity. We reconstructed the patterns of sample-evoked MEG responses within well-defined cortical regions (Methods)[13] and quantified the information about task-relevant variables contained in these neural responses (Fig. 3). We performed comprehensive mapping of information measures across the cortical surface (Fig. 3b, d, and f) complemented by analyses focusing on "Dorsal Stream Visual Cortex"[22], a group of visual field maps in posterior parietal cortex that is located within and around the intraparietal sulcus[23,24] (henceforth referred to as "dorsal visual cortex"; Methods) to precisely track the dynamics of information measures (Fig. 3c, e, and g; see Supplementary Fig. 6 for event-related MEG responses across the complete trial). We chose parietal cortex, and dorsal visual cortex in particular, as a primary region of interest because of its established role in visual-spatial behavior[25–27].

We quantified three information-theoretic measures (whose relationship is schematized as a Venn diagram in Fig. 3a, right): (i) I(S;R), the information about the sensory evidence sample S encoded by the neural population response R; (ii) I(R;E), the information about the upcoming behavioral (estimation) report E encoded by R; and, critically, (iii) the intersection information II(S;R;E), a statistical measure of how much of the sensory evidence sample information encoded in R is read out to inform the behavioral estimation report E[18]. Figure 3 depicts those measures, as time courses following the onset of each sample (in dorsal visual cortex) and as maps across all areas (pooled across samples), for the trial interval of interest that followed the categorical choice or cue.

Our analyses yielded spatio-temporal profiles of information measures that were in line with the profiles obtained in decoding approaches in similar tasks[4,13]. I(S;R) peaked ~150 ms in many visual cortical areas after sample onset (Fig. 3c). I(R;E), the information about the upcoming estimation report (Fig. 3d, e) was likewise present in many cortical areas, here in a more sustained fashion (Fig. 3e), because it built up gradually throughout the processing of all evidence samples. In the Choice condition, I(R;E), was also prominent in frontal and parietal cortical regions involved in action planning (Fig. 3d). Dorsal visual cortex exhibited robust I(S;R) and I(R;E) (Fig. 3b–e), corroborating our focus on this brain region. The profile of intersection information (measure of readout of encoded evidence) II(S;R;E) exhibited

spatio-temporal features resembling those of both I(S;R) and I(R;E) again with a robust effect in dorsal visual cortex (Fig. 3f, g). Taken together, our results are in line with the idea that the sample information encoded in dorsal visual cortex was read out for behavioral estimation reports (Fig. 3).

### Readout of stimulus information depends on consistency with previous choice or cue

In order to illuminate the neural basis of the effect of evidence consistency with the preceding choice on the final estimate, we compared the neural information measures between evidence samples that were consistent with the intermittent choice (or cue), and those that were inconsistent (Fig. 4). Because we found behavioral consistency effects on I(S;E) for samples 7 and 8 (Fig. 2), we combined both sample positions for our analyses of neural information measures to increase the minimum trial counts available after splitting by sample consistency (Methods). Consistency-dependent evidence encoding would predict higher I(S;R) for consistent than inconsistent samples. The data were inconsistent with this hypothesis: for both Cue and Choice conditions, we found no significant difference in I(S;R) between consistent and inconsistent samples in dorsal visual cortex (Fig. 4a, b; see Supplementary Fig. 7 for effect maps for other trial intervals). The Bayes factor (Methods) provided strong support for the null hypothesis (BF = 0.210) in the Cue condition (Fig. 4a). In the Choice condition, the pattern of I(S;R) even tended to be flipped (inconsistent > consistent), with weak support for the null hypothesis (BF = 0.359); Fig. 4b.

In contrast, intersection information II(S;R;E) was significantly larger in dorsal visual cortex for consistent than for inconsistent samples (Fig. 4e, g), an effect evident in the first principal component alone (Supplementary Fig. 8). In line with the effect of consistency on behavioral evidence weighting (Fig. 2), the effect of consistency on II(S;R;E) was temporally specific for sample positions 7 and 8, with no significant effects earlier in the trial (see Supplementary Fig. 9 for first sample positions and Supplementary Fig. 10 for samples immediately preceding the cue or choice). Dorsal visual cortex exhibited the strongest consistency effect on II(S;R;E) across the entire cortex in both Cue and Choice conditions (Fig. 4f, h), but the consistency effect was also expressed (and statistically significant after FDR correction across regions) in a neighboring area of posterior parietal cortex, the "Inferior Parietal Cortex" group[22] (Fig. 4f, h).

The cortex-wide spatial patterns of the consistency effect on neural II(S;R;E) were similar between Cue and Choice conditions (compare maps in Fig. 4f and h; correlation coefficient: 061). We also evaluated this pattern similarity at a finer level of granularity (using 180 individual areas instead of the 22 area groups[22]) combined with rigorous spatial-autocorrelation preserving permutation tests (Methods). Also this conservative approach showed significant correlation between the spatial maps of consistency effect on II(S;R;E) in Cue and Choice (Fig. 4j). No significant correlations were found for the maps of the consistent-inconsistent difference in I(S;R) or I(R;E) (Fig. 4i), highlighting the specificity of the pattern similarity shown in Fig. 4j for the effects of consistency on neural intersection information.

Our results are in line with the notion that the readout of the sample information in parietal cortex for the estimation report depended on the sample's consistency with the previous categorical choice or cue. Such an effect of evidence consistency on II(S;R;E) can be understood from changes in the geometry of neural codes in parietal cortex (see schematic in Fig. 4k and model simulations in Supplementary Fig. 5). For simplicity, we consider a two-dimensional neural state space made up of two neural features (e.g., the activity levels of two neural populations or of two principal components of the local activity pattern). Within this two-dimensional subspace of neural activity, the two vectors represent the neural activity patterns encoding stimulus position and behavioral estimation reports. The alignment between these vectors, in turn, determines how efficiently the

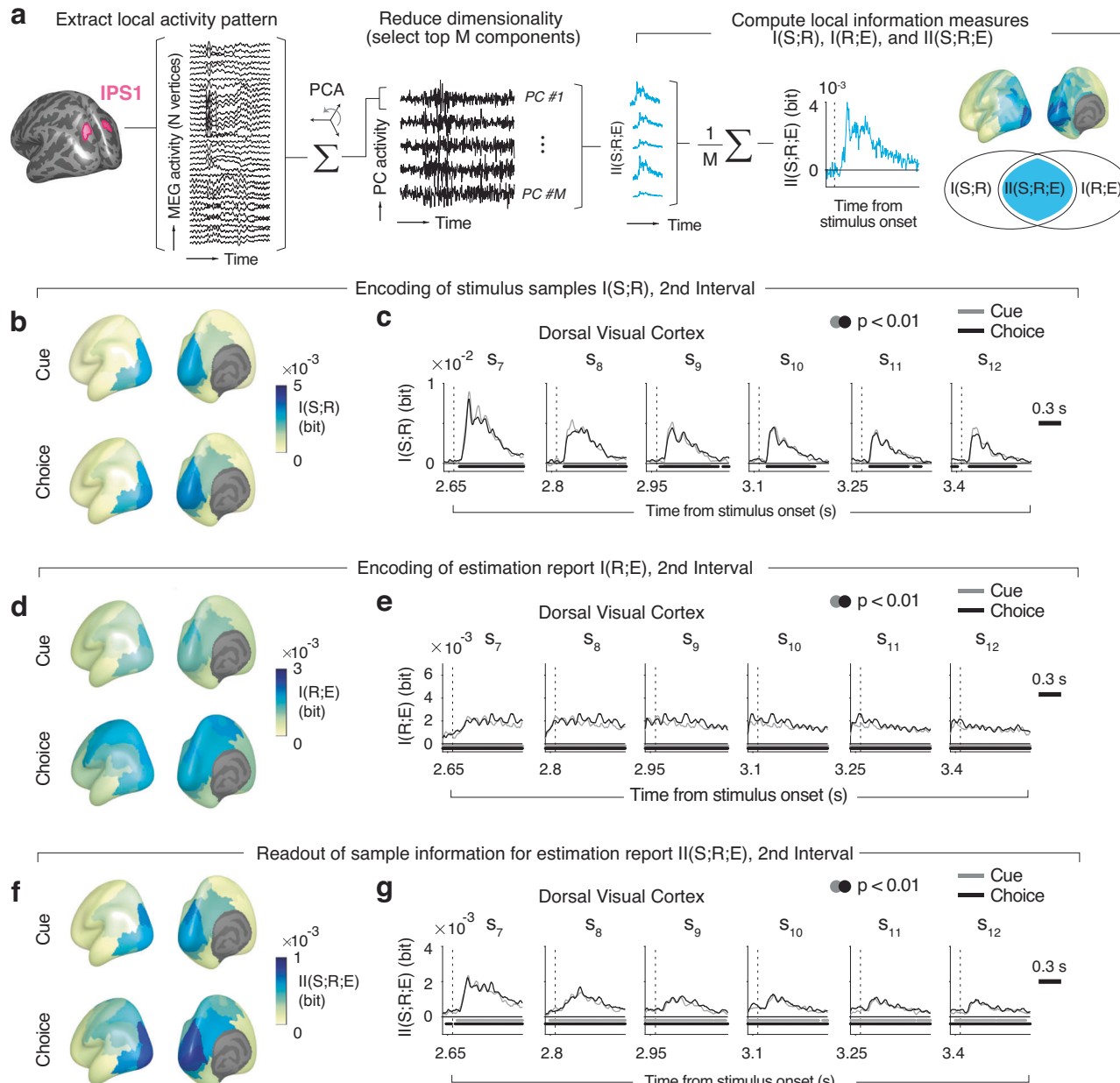

**Fig. 3 | Encoding of evidence and estimation report in cortical population activity. a** Schematic of analysis approach. The dimensionality of source-reconstructed neural activity of all vertices per cortical area (example: IPS1 from dorsal visual cortex) was reduced with principal component analysis (PCA). *M* top components (between 1 and 15) that jointly explained 90 % of the variance were selected for all (information theoretic and decoding) analyses. Stimulus sample information I(S;R), estimation information I(R;E), and intersection information II(S;R;E) were computed for each component and then averaged (see main text for details). **b** Maps of group average I(S;R) following sample onset (averaged across samples from second interval, positions 7–12). **c** Time course of I(S;R) in dorsal visual cortex for each sample position of second interval. **d, e** As in (**b, c**), but for I(R;E). **f, g** As (**b, c**), but for II(S;R;E). For all maps, we averaged information measures across the interval 0.1 s–0.5 s after sample onset, which contained most of the stimulus information (see panel c). Time courses were smoothed with a Gaussian kernel (sigma = 37.5 ms), and baseline-corrected with the mean information in a −0.1 s – 0 s window before the sequence onset. Bars beneath time courses indicate time points significantly different from baseline (*p* < 0.01, cluster-based two-sided permutation tests) for Choice (black) or Cue (gray). *N* = 30 participants for all panels.

stimulus information encoded in neural activity is read out to inform behavioral reports[17]. Mechanistically, a change in the match for a fixed geometry of the stimulus representation can result from a change in the pattern of weights applied to that representation in the decision process[28]. In this scheme, the effect of evidence consistency on behavior (I(S;E); Fig. 2) and on II(S;R;E) in parietal cortex (Fig. 4e–h) can be explained by a mismatch between the stimulus and estimation vectors for inconsistent samples compared to consistent samples. (Fig. 4k). This explains why equally strong stimulus information

(Fig. 4a, b) does not propagate to behavior when a sample is inconsistent.

To further support this explanation and test our model predictions (Supplementary Fig. 5), we developed an analysis that approximates intersection information within the context of linear regression. To quantify how well sample information decoded from neural population activity was predictive of behavior, we trained linear decoders to predict the sample stimuli from the neural data in each area and then correlated the decoder predictions, on separate data,

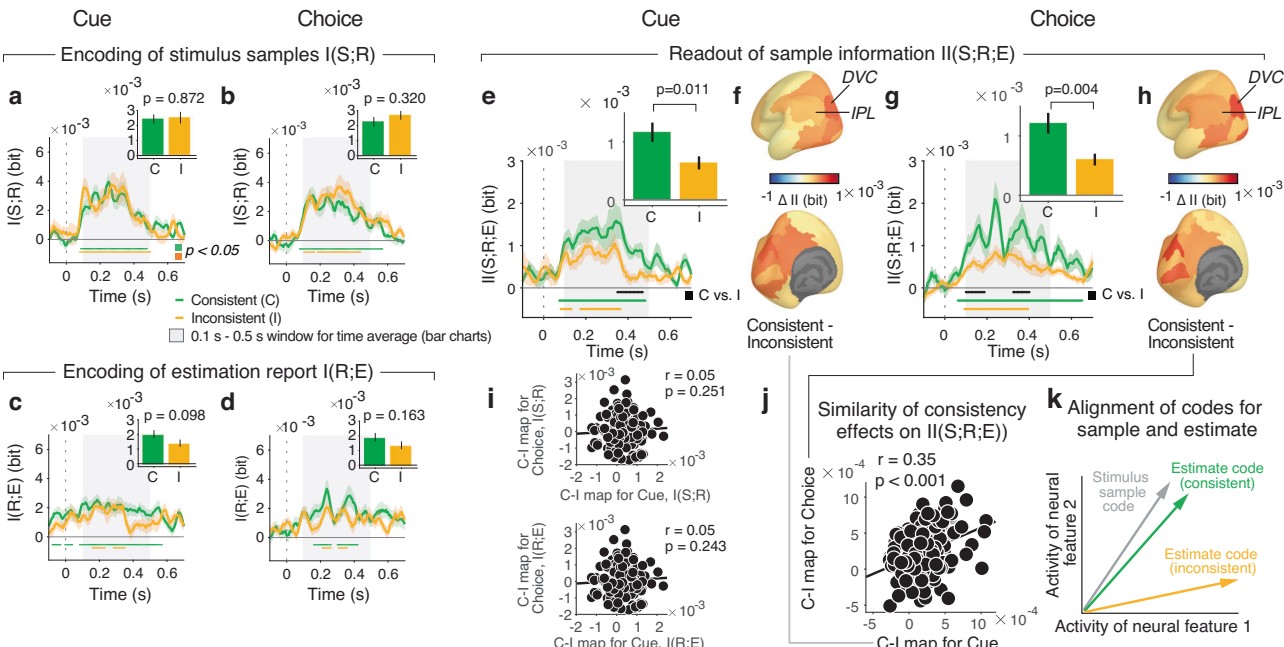

**Fig. 4 | Effect of consistency on evidence encoding and evidence readout.**
**a, b** Time courses of stimulus sample information I(S;R) in dorsal visual cortex for consistent (C) and inconsistent (I) samples, relative to cue (**a**) or choice (**b**). Time-courses are pooled across samples 7 and 8, showing the behavioral consistency effect in Fig. 2. Bar charts, mean across gray-shaded interval with significant I(S;R). Bars or lines, group mean; error bars or shaded areas, SEM. Horizontal bars, significant difference from (−0.1 s to 0 s) baseline interval (colors) or between consistent and inconsistent (black); *p*-values from two-sided permutation tests (cluster-corrected for time courses); *N* = 29 participants. Time courses were smoothed with Gaussian kernel (sigma = 37.5 ms). **c, d** As (**a, b**) but for estimation information I(R;E). **e, g** As (**a, b**) but for intersection information II(S;R;E). **f, h** Maps of consistency effects on II(S;R;E) displayed without threshold. Abbreviations: DVC, dorsal visual cortex; IPL, inferior parietal cortex. **i** Correlation between maps of

consistency effects on I(S;R) and I(R;E) in Cue and Choice conditions (Pearson's correlation coefficient, one-sided). Data points, areas (*N* = 180); *p*-values from spatial autocorrelation-preserving permutation tests (Methods). **j** As (**i**) but for pattern similarity of consistency effect maps on II(S;R;E). **k** Schematic of mismatch between neural stimulus encoding and readout. Vectors represent neural codes for sample stimuli (gray) or behavioral estimates (colors) in a two-dimensional state space. The alignment of the neural activity patterns governs the contribution of encoded stimulus information to behavioral report. When samples are consistent, the readout is well-matched (green vector). When samples are inconsistent, the readout is mismatched (orange vector). Also the stimulus vector may change between conditions; we show a single stimulus vector for simplicity: What matters is the relative alignment of the codes for stimulus and estimate within condition. See Supplementary Fig. 5 for simulations.

with participants' estimates (Methods). This approach yielded results similar to the information-theoretic analyses, again showing indistinguishable encoding of consistent and inconsistent evidence, but a significant effect of consistency on evidence readout (Supplementary Fig. 11).

**Linked individual consistency effects in behavior and readout from parietal cortex**

If the effect of consistency on intersection information in parietal cortex (Fig. 4) contributed to the consistency-dependent evidence weighting profile (Fig. 2), then individual differences in the strengths of these neural and behavioral consistency effects should be related. Having established similar behavioral and neural consistency effects for Choice and Cue conditions, we correlated participants' condition-averaged consistency effect on II(S;R;E) (Fig. 5b) with the corresponding consistency effect on I(S;E) (Fig. 5a). These effects were correlated across participants for the two posterior parietal regions showing robust consistency effects on intersection information: dorsal visual cortex and inferior parietal cortex: most strongly (across the entire cortex) for inferior parietal cortex (Fig. 5c–e). This correlation was negligible when using I(S;R) instead of II(S;R;E) as neural measure (Fig. 5f), again highlighting the specificity for intersection information. Together with Fig. 4, the results from Fig. 5 are consistent with the idea that the bias of evidence processing induced by an intermittent choice or cue is mediated by an altered readout of that evidence from parietal cortex: Evidence following the choice is read out from parietal cortex

in a manner that depends on consistency, and the individual strength of this consistency effect reflects the individual strength of consistency effect on evidence weighting observed in behavior.

## Discussion

Human judgment and decision-making are shaped by biases and heuristics[29–31]. One of the most prevalent of those is the tendency to process new evidence selectively, in order to confirm the beliefs and judgments one has previously committed to[7,9,32], whereby evidence supporting a previous decision is given more weight in future judgments compared to contradicting evidence[8,31]. We reasoned, and confirmed in model simulations, that such selective processing of post-decisional evidence could be caused either by a change in the encoding precision of evidence in sensory cortex, or a change in the readout of that sensory representation for subsequent decisions. Our information-theoretic dissection of evidence encoding and readout in patterns of human cortical population activity provide no evidence for the encoding scenario. Instead, our results clearly support the readout scenario and implicate posterior parietal cortex as an important processing stage in the underlying neural pathway.

This result is important for unraveling the information processing mechanisms underlying choice-induced biases. Choice-induced biases can be advantageous under certain conditions[33] but may be detrimental in others. Maintaining sensory representations that are undistorted by categorical beliefs is beneficial for the observer. It can prevent the erroneous maintenance of beliefs states (hysteresis) when

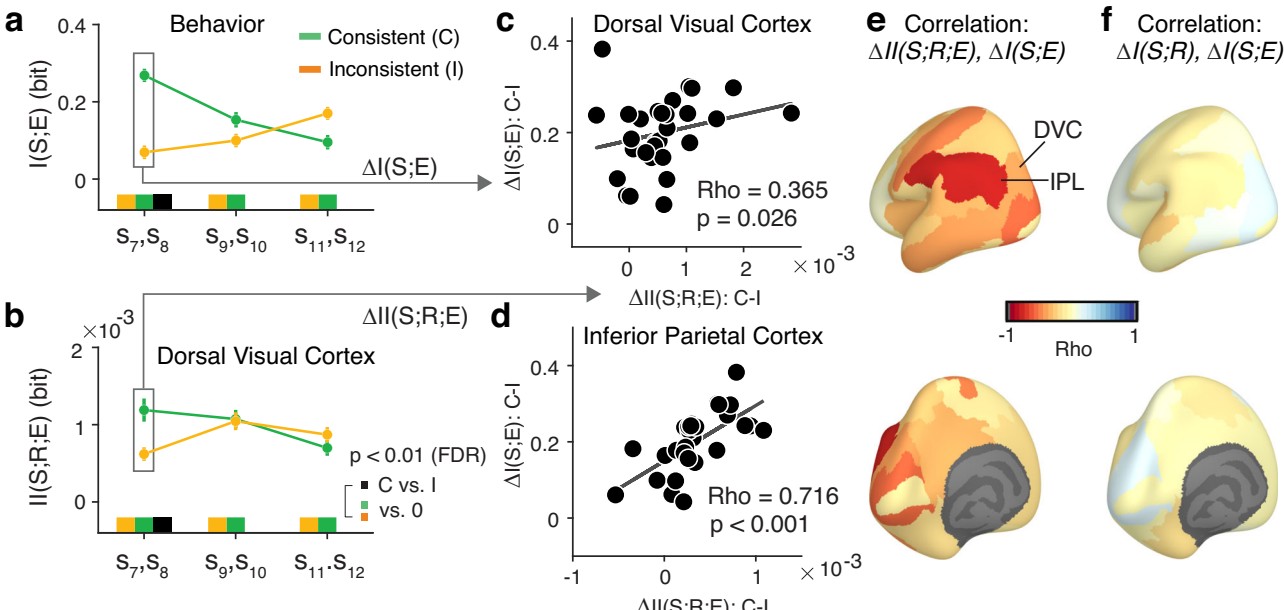

**Fig. 5 | Individual differences in neural and behavioral consistency effects are related. a** Behavioral consistency effect as difference in psychophysical kernels I(S;E) between consistent and inconsistent samples, pooled for two samples in second trial interval, averaged across Cue and Choice conditions. **b** Corresponding neural consistency effect as difference in intersection information II(S;R;E) between consistent and inconsistent samples in dorsal visual cortex (0.1 s−0.5 s from sample-onset, corresponding to bar graphs from Fig. 4e, g). Marks below time courses, statistical significance (two-sided permutation tests, FDR-corrected). **c, d** Across-

participant correlations between behavioral and neural consistency effects for samples 7,8 (box in panels a,b) in dorsal visual cortex (**c**) and inferior parietal cortex (**d**), averaged across Choice and Cue conditions. Numbers, Spearman Rho and associated *p*-values (one-sided). **e** Map of across-participant correlations (Spearman's Rho) of behavioral versus neural consistency effects for II(S;R;E). Abbreviations: DVC, dorsal visual cortex; IPL, inferior parietal cortex. **f** As (**e**) but substituting stimulus information I(S;R) for II(S;R;E) as neural measure. N = 29 participants for all panels; data points, group mean; error bars, SEM.

additional evidence or an actual change in source location warrants a switch in categorical belief[4]. Indeed, behavioral findings on retrospectively (i.e., after evidence presentation) induced choice biases show that perceptual estimation behavior after a forced change in choice due to feedback is best modeled by a switch in categorical belief, but unaltered sensory representations[34], in line with our current neural data. More generally, pinpointing the origin of confirmation bias to a modulation of evidence readout rather than evidence encoding suggests that confirmation biases may result from a deliberate decision process, and thus can potentially be counter-balanced by appropriate incentives.

Other work has identified post-decisional biases in other perceptual designs that also entail categorical and continuous estimation judgments, but, differently from our task, with both judgments prompted after termination of the evidence stream[8,19,35]. The biases in the continuous estimation that are here "retrospectively" induced by the categorical choice resemble established post-decisional biases in value-based decisions[36]. These results point to self-consistency as a general principle shaping human decision-making[19,37]. Here, we found that such choice-induced consistency effects also act "prospectively" and pinpointed the readout of sensory evidence as the source of the biased processing. Whether or not this mechanism also underlies the retrospective post-decision bias described above remains to be tested in future work.

We used two complementary approaches to modulate participants' categorical beliefs about the source in the middle of the evidence stream: prompting them to report a judgment about that category or providing an external cue that informed about that category in a probabilistic manner. Both manipulations induced a similar bias in subsequent evidence weighting and neural readout (intersection information). The consistency effect on behavioral evidence weighting was stronger in the Choice context, suggesting that this

effect is amplified when the bias-inducing event is self-generated (Choice) compared to when it is externally provided (Cue). Cue- or choice-related biases in the overall estimation reports were clearly present in both task contexts (Fig. 1h), albeit with an opposite dependence on the strength of category evidence (Fig. 1i). Similar biases in the final estimates for both contexts are expected from the short-lived nature of the differential consistency effects between conditions observed in the psychophysical kernels (Fig. 2c). These consistency effects (and their difference between Choice and Cue) were present in two samples following the categorical events, thus affecting the integration of a small fraction of the total evidence available for the final estimation decision. Our precise psychophysical and neurophysiological tools for dissecting evidence encoding and readout at high temporal precision enabled us to identify such temporally specific modulations of the readout. It will be interesting to test whether the relative impact of interim choices or cues on final estimates indeed increases when those categorical events are followed by fewer evidence samples.

Our results suggest that the choice-induced bias characterized here is unlikely to be caused by a selective modulation of sensory representations[38–40], for example through feedback signals[41–43]. We found no evidence for a change of stimulus encoding in visual cortex, depending on the stimulus consistency with the previous choice or cue. Our manipulation of categorical beliefs with the external cue invites comparison with a large body of studies using cues to manipulate selective attention and measured neural activity in visual cortex. Many of those studies reported changes in the sensory representation via gain modulation[38]. This apparent discrepancy may be due to differences between our current approach and the previous studies of selective attention. For example, we presented imprecise cues that divided the continuous visual feature space into two coarse categories (left vs. right from reference line), whereas the cues have commonly

been precise in most previous studies on attention mechanisms. Furthermore, we quantified sensory encoding in terms of the mutual information between stimulus and an MEG-derived estimate of the local field potential response, as opposed to single neuron firing rates in the previous studies. Finally, it is possible that the reported effects of attention on sensory encoding precision co-existed with stronger effects on sensory readout, which largely mediated the behavioral effects of attention[44]. It would be useful to apply our approach to a wider range of cueing tasks including those commonly used in the attention literature.

The interplay between encoding and readout of information, operationalized here as intersection information, has been identified as an important criterion for identifying neural activity patterns guiding perceptual decisions[17] (see also[45–47]). The rationale is that not all information about the stimulus encoded in the brain necessarily informs behavior; likewise, not all neural activity shaping behavior is based on the readout of stimulus information (e.g., choice history biases[48] or motor biases). Because intersection information quantifies the information about a stimulus contained in a neural activity pattern that informs behavior, it enables the most specific statistical (i.e., correlative) delineation of putative task-relevant neural signals in the brain. Here, intersection information enabled us to isolate a clear effect of consistency that directly originates from the sample evidence that was neither evident in stimulus nor estimation information. Given the limits of recordable data, we averaged all information quantities across features (i.e., principal components) of the neural activity pattern within a given area, rather than evaluating the joint information across neural features. Thus, while unlikely, we cannot exclude the possibility that there was a consistency effect in the joint stimulus information across neural features.

A few previous studies have probed the neural basis of confirmation biases[9,32]. One reported behavioral (drift diffusion) model fits indicating that sensory evidence following an initial categorical judgment of a noisy stimulus was selectively accumulated depending on its consistency with the initial judgment, in a manner that depended on the observers' confidence about the initial judgment, a conclusion that was supported by the dynamics of decision-related (sensor-level) MEG signals[9]. Selective evidence accumulation, however, does not allow to distinguish between a selective modulation of evidence encoding and a selective evidence readout[49]. In a social setting, fMRI responses in the posterior medial prefrontal cortex indicated failure to use other's opinion when they were disconfirmatory[32]. None of these studies have pinpointed the neural computations underlying selective evidence integration, which did here by distinguishing between the encoding versus the readout of evidence for decisions.

Our findings resemble those from studies of the mechanisms of visual perceptual learning in macaque monkey cortex[28,50]. This work uncovered an improvement of the readout of the sensory representation from visual motion-selective cortical area MT by downstream decision circuits during learning[50], an effect that could be modeled as a reinforcement-dependent optimization of readout weights[28]. Our current work establishes a similar change in sensory readout weights, but on a substantially more rapid timescale, at the level of trial-to-trial and even sample-to-sample (within trial) resolution. This analogy points to the possibility of systematically changing individuals' tendencies toward self-confirmation through directed learning incentives.

In conclusion, adopting categorical beliefs about the state of the environment does not affect the encoding of subsequent evidence in the human brain, but selectively biases the readout of the encoded information for future decisions. Because confirmation bias is prevalent in cases of real-world significance, our insights have important practical implications. For example, evidence readout is more susceptible to strategic interventions and learning than evidence

encoding. Our results indicate that the room for manipulating self-confirmation tendencies in real-life decisions may be substantial.

## Methods

### Participants

34 healthy human volunteers (sex: 16 male, 18 female) aged between 19–39 years (mean = 27 years, SD = 5) participated in the study. All participants gave written consent and were naive to the objectives of the study. Participants self-reported normal or corrected-to-normal vision and no history of psychiatric or neurological diagnosis. They received remuneration in the form of an hourly rate of 10 €, and a bonus (25 €) for completing all planned sessions. The study was approved by the ethics review board of the Hamburg Medical Association responsible for the University Medical Center Hamburg-Eppendorf (UKE).

No power analysis was used prior to the study, because the effects of interest (dependence of stimulus encoding or readout in the brain) are unknown. We based our sample size of $N = 34$ on those from previous MEG experiments using comparable stimulus and task designs[4,13]. The chosen sample size was larger than the ones these papers, and we collected a large amount of data from each of the 34 participants in our experiment (at least 1856 trials).

Four participants were excluded during the MEG preprocessing or source reconstruction stage (i.e., prior to the tests of the main experimental effects, see below), leaving $N = 30$ participants. Three of those showed persistent MEG artifacts (of unclear origin) at the sensor-level, which could not be removed with the preprocessing pipeline described below; in a fourth, the source reconstruction (see below) failed for several parcels. One additional participant did not have a sufficient trial count for quantifying neural intersection information separately for consistent and inconsistent conditions (see below, Information-theoretic analyses of behavior and neural activity.), leaving $N = 29$ for Figs. 4 and 5 and associated Supplementary Figures on neural consistency effects on information measures. The behavioral consistency effects (i.e., on I(S;E)) shown in Fig. 2 were unaffected when restricting the analysis to the same $N = 29$ participants included in Figs. 4 and 5.

### Stimuli

Stimuli were generated using Psychtoolbox-3 for MATLAB and were back-projected on a transparent screen using a Sanyo PLC-XP51 projector at 60 Hz during MEG recordings, or on a VIEWPixx monitor during the training session in a behavioral psychophysics laboratory. The task-relevant stimulus, so-called evidence samples, were small, checkered discs (0.8° diameter, temporal frequency, 6.7 Hz; spatial frequency, 2°) whose eccentricity was fixed at 5° of visual angle and whose polar angle varied from sample to sample within each trial. Each sample was presented for 100 ms followed by 50 ms blank. Two placeholders were present throughout each trial: a fixation cross at the center of the screen with varying orientation and color informing participants about trial intervals, and a light-gray reference line extending downward (or upward) from fixation to 5° eccentricity (Fig. 1a). In different trial blocks, the sample positions were restricted to the upper or lower hemifield (counterbalanced). The orientation of the checkerboard pattern changed according to the sample position. Stimuli were presented against a gray background.

### Definition of sensory evidence

We defined sensory evidence as sample positions with respect to the reference line: for both upper hemifield blocks and lower hemifield blocks, a sample position of 0° indicated that discs were positioned on the vertical meridian (i.e., aligned with the reference line), positive angles indicated locations right from the reference line, and negative angles indicated locations left from the reference line. In each trial, the sample positions were drawn from a Gaussian distribution, with a

(generative) mean that varied from trial to trial and fixed standard deviation of 20°. Generative means for each trial were sampled from a uniform distribution (−14°, +14° relative to reference line in 1° increments −15°, +15°, in 5° increments for the first 3 participants). All samples were constrained to the range of −40° to + 40° from the generative mean.

## Behavioral tasks

Participants were asked to monitor a total of 12 evidence samples, infer their hidden source (i.e., generative mean), and report that as a continuous estimate at the end of the trial. After the first six samples, participants were also asked to report a categorical judgment of the evidence source (left or right from reference line; Choice condition); or to make a stereotypical button press that was pre-assigned for each block of trials (Cue condition). In the latter task, the button press was followed by an imprecise visual cue indicating the most likely (75% validity) category of the evidence source. This cue was as an arc spanning 14° of polar angle (i.e., the range of generative means on either side of the reference line). Participants were instructed to maintain fixation throughout the trial.

Each trial started with a baseline interval ranging between 0.5 s – 1 s. The intermittent response time window was fixed at 1.25 s, the cue display period was fixed at 0.5 s immediately after the response window. The offset of the final sample was followed by a delay interval of 0.1 s–0.3 s, during which the fixation cross was replaced by an 'x'. After that delay, the reference line turned red, which was the go-cue prompting the participant's estimation report. The time window for production of the estimate ended with the participants' report or at 5 s, whichever came first. The inter-trial-intervals were drawn from a uniform distribution (3.5 s, 5 s). Cumulative feedback about the binary choice (percentage correct; if the generative mean was zero, the correctness was assigned based on randomly choosing left/right as correct), and the estimation (root mean squared deviation of estimation from the generative mean) were provided at the end of each block. In addition, feedback about the number of trials they have missed the button press or estimation, as well as the number of trials in which they have failed to maintain fixation was also provided.

## Experimental procedure

All participants completed five sessions on separate days: one behavioral training session in a psychophysics laboratory where participants were instructed about, and familiarized with, the task; and four main experimental sessions in an MEG laboratory.

**Behavioral training session.** On the first visit, participants were briefed about the purpose of the study and measurement. Since the measurements took place during the COVID-19 pandemic, participants submitted negative tests and were informed about additional hygiene measures implemented to minimize risk from the virus. Experimenters also were required to test negative before any measurements during this period. For the training, participants performed a behavior-only version of the main task broken down into seven steps, with their head fixed on a chinrest, 60 cm from the monitor. The training started with practicing making a left/right choice by accumulating the 12 checkered dot positions presented sequentially. The final step of the training was similar to the main task except that the intermittent button press was preassigned and there was no cue. The training lasted around 2 h. Participants were trained both with and without feedback about their performances on a trial-by-trial basis. Participants' performance was checked immediately after data collections.

**MEG sessions.** Participants performed two sessions of the Cue task, followed by two sessions of the Choice task, in that order, on separate days. We used this fixed order to prevent participants from forming

internal categorical judgments also in the Cue condition. Each session consisted of 8 blocks (58 trials each). In addition, visual cortex localizer blocks were added at the end of each session for both upper and lower hemifield stimuli. Participants were seated 61 cm from the screen during MEG recording. Before each session, participants underwent a brief practice block to familiarize themselves with the joystick control in the MEG environment. Each session contained around 2 h of recording time, resulting in a total of ~6-h measurement per task. Identical trials were used for both tasks and all participants. In addition, there were 4 identical trials for each session, 3 in the exact sequential order, and one shuffled across the 12 sample positions. We collected 1856 trials (928 trials per task) in total from each participant during MEG.

## Data acquisition

MEG data were acquired with 275 axial gradiometers (equipment and acquisition software, CTF Systems) in a magnetically shielded room (sampling rate, 1200 Hz). Participants were asked to minimize any movements. Head location was recorded in real-time using fiducial coils at the nasal bridge and each ear canal. A template head position was registered at the beginning of the first session, and the participant was guided back into that position before each subsequent task block. Ag–AgCl electrodes measured the ECG and EOG which was monitored throughout the session. Eye movements and pupil diameter were recorded at 1000 Hz with an EyeLink 1000 Long Range Mount system (equipment and software, SR Research). On a separate day, T1-weighted MRIs were acquired to generate individual head models for source reconstruction. Behavioral data (button presses and joystick displacement for estimation reports) were collected simultaneously with the electrophysiological data.

## Data analysis

Data were analyzed in MATLAB (MathWorks) and Python using a combination of custom code (see Code Availability statement below) and the following toolboxes: FieldTrip[51] (https://www.fieldtriptoolbox.org), MNE-Python[52] (https://mne.tools/stable/index.html), MINT[53] (https://github.com/panzerilab/MINT), and pymeg (https://github.com/DonnerLab/pymeg).

**Trial exclusion.** Some trials were excluded from analysis based on criteria related to behavior and MEG data. Trials with missed intermittent button press (choice condition) or missed final estimation were excluded. Also, trials with final estimation reaction time shorter than 300 ms were excluded, since these trials were deemed too fast to have made a decision, considering the fact that the estimation cursor was always located aligned to the vertical reference line. More specifically, trials that could induce a zero generative mean bias were excluded. Based on MEG data, trials with the following artifacts were excluded: sensor noise, muscle artifacts, eye-movement related artifacts (blinks, saccades). Trials used in both behavioral and MEG analyses were matched. On average, 22.1% of trials (standard deviation: ±10.7%) were excluded per individual dataset, leaving 723±99 (mean and standard deviation) trials per participant for the analyses described below.

**Analysis of choice and estimation reports.** Performance of the intermittent categorical judgment (Choice task only) was evaluated by fitting a cumulative Gaussian as a function of the sample mean for the first 6 samples. We pooled estimation reports across all sessions of a given participant, normalized the values, and then pooled the normalized values across participants. We used Hartigan's dip test to test for deviations from unimodality of the estimation distributions. We also computed and compared the mean estimation reports after sorting based on the category (i.e., left or right) of the choice or the cue, separately for three bins of category evidence strength.

**Ideal observer.** As a reference for the interpretation of observed behavioral effects, we also simulated a simple noise-free and unbiased ideal observer model on the exact same sequences of evidence seen by the participants. The ideal observer made intermittent choices based on the sign of the mean of samples 1-6, and continuous estimations based on the mean of all 12 samples (i.e., "perfect integrator"). The behavior of this ideal observer was analyzed in the same way as the behavior of the participants.

**MEG preprocessing.** Continuous data were segmented into task blocks, high-pass filtered (zero phase, forward-pass FIR) at 0.1 Hz and band-stop filtered (two-pass Butterworth) around 50 Hz, 100 Hz and 150 Hz to remove line noise. The data was down-sampled to 400 Hz. Trials containing the following artifacts were discarded: translation of any fiducial coil > 6 mm from the first trial, blinks (detected by the EyeLink algorithm), saccades > 1.5° amplitude (velocity threshold, 30° s$^{-1}$; acceleration threshold, 2,000° s$^{-2}$), squid jumps (detected by Grubb's outlier test for intercepts of lines fitted to single-trial log-power spectra), sensors with data range > 7.5 pT and muscle signals (after applying a 110 Hz – 140 Hz Butterworth filter and z scoring) of z > 10. In addition, trials containing artifacts due to cars passing by in the vicinity of the building were detected and discarded. The remaining data for that recording were then subjected to temporal independent component analysis (infomax algorithm), and components containing blink or heartbeat artifacts were identified manually and removed.

**Source reconstruction.** We used linearly constrained minimum variance (LCMV) beamforming to estimate activity time courses at the level of cortical sources. For source reconstruction, data were segmented into epochs ranging from 0.5 s before 1$^{st}$ sample onset and 4.5 s after, which includes the two evidence streams and the intermittent button press (and cue, for Cue condition). We constructed individual three-layer head models (except for three participants, for whom we used one-layer models) from structural MRI scans using FieldTrip. Head models were aligned to the MEG data by a session-specific transformation matrix generated with MNE. Transformation matrices were generated using MNE software. We reconstructed individual cortical surfaces from MRIs using FreeSurfer and applied the parcellation from an aligned anatomical atlas[22] to each surface. We used MNE to compute LCMV filters confined to the cortical surface (4,096 vertices per hemisphere, recursively subdivided octahedron) from the covariance matrix computed across all time points of the cleaned and segmented data. The covariance matrix was averaged across all trials per session. For each vertex location, we determined the source orientation yielding maximum power by means of singular value decomposition. The LCMV filters were used to project single-trial broadband MEG time series into source space. We computed MEG responses at each vertex by first aligning the polarity of time series at neighboring vertices, since the beamformer output potentially included arbitrary sign flips for different vertices.

**Parcellation and regions of interest (ROIs).** We used the atlas from Glasser and colleagues[22] as parcellation of the cortical surface into 180 well-defined areas based on multimodal (functional and anatomical) data. All MEG data analyses described below were first performed separately for each of these 180 areas. Because the dimensionality of source-level MEG data is typically smaller than 180, we averaged the resulting information theoretic measures across all areas from each of 22 area groups described in reference[22]. This was done for all analyses except the spatial pattern correlations described below. Complementary to the comprehensive mapping of effects across all the entire cortex, our analyses focused on two posterior parietal area groups as specific ROIs: dorsal visual cortex and inferior parietal cortex. We chose posterior parietal cortex, and "Dorsal Stream Visual

Cortex" (referred to as "dorsal visual cortex" for brevity)[22] as a priori ROI due to its established role in visual-spatial behavior[25–27] for all figures showing neural analyses. Dorsal visual cortex is a group of visual field maps within and around the intraparietal sulcus, comprising the following areas: V3A, V3B, V6, V6A, V7, and IPS[23,24]. To further depict correlations between individual effects of consistency on behavioral evidence weighting versus neural intersection information, we also chose a neighboring parietal region, the "Inferior Parietal Cortex" group (comprising areas PGp, IP0-2, PF, PFt, PFop, PFm, PGi, PGs) as an additional ROI. This selection was based on the significant consistency effect on intersection information (Fig. 4) in that group.

**Dimensionality reduction.** We used principal component analysis (PCA) across all vertices and trials, within each parcel to reduce the dimensionality of the data within each of the 180 parcels The source data was down-sampled to 200 Hz (for Figs. 3, S6, and S12) or 160 Hz (for all other figures showing neural data). This resulted in a 1001 sample segment for a trial (−0.5 s ~ 4.5 s; 0 s: 1$^{st}$ sample onset). We then chose $M$ principal components which cumulatively explained 90 % of the variance in the data. The number of selected principal components for each parcel ranged from 1–15. All analyses reported in the following were done on these principal components.

**Information-theoretic analyses of behavior and neural activity.** We used mutual information to quantify the amount of information encoded in neural activity ($R$) about stimulus samples $S$ or participants' behavioral estimation reports $E$, as a measure of sensory evidence (or report) encoding in neural activity. Stimulus samples $S$ and behavioral estimation reports $E$ were both expressed as angular positions with respect to reference line (see *Definition of sensory evidence* above). Mutual information quantifies the effect of all the single-trial (both linear and nonlinear) statistical relationship between two random variables[54,55]. For each parcel (cortical area) and task variable $X$ ($S$ or $E$), we computed mutual information carried by $R_{i,t}$, that is, the activity of each principal component $i$ (see above, *Dimensionality reduction*) and each time point $t$, as follows:

$$I(X;R_{i,t}) = \sum_{x,r_{i,t}} p(x,r_{i,t})\log_2 \frac{p(x,r_{i,t})}{p(x)p(r_{i,t})} \qquad (1)$$

where $p(x,r_{i,t})$ was the joint probability of observing a neural or behavioral response $r_{i,t}$ and for a value $x$ of the task-relevant variable, and $p(r_{i,t})$ and $p(x)$ were the marginal probabilities of $r_{i,t}$ and $x$ respectively. The sum is over all possible values of $r_{i,t}$ and $x$. We discretized the values of neural responses $R_{i,t}$ (independently for each time point and PC) and behavioral variables $X$ (both $S$ and $E$ were continuous) across trials into equally populated bins[56]. Equally populated binning is advantageous for calculations of information about continuous-valued variables because it maximizes the marginal entropy of the variables given the number of bins and this often increases the mutual information values[56]. We then estimated $p(x,r_{i,t})$ in Eq. (1) from data using the direct method[57,58] (i.e., counting the occurrences of each possible pair of $(r_{i,t},x)$ values across trials and dividing by the total number of trials). Specifically, we quantified the relationship between trial-to-trial fluctuations in the stimulus $S$ at each sample position in the sequence and the final behavioral estimation reports $E$ by computing I(S;E). The resulting psychophysical kernels measure the time course of the impact of stochastic fluctuations in sensory evidence samples on behavioral reports[21].

We used intersection information (II)[18] to quantify the amount of stimulus information encoded about the stimulus $S$ by the neural signal $R_{i,t}$ that informed the behavioral estimation report $E$. This measure is computed from the observing probabilities of stimulus $S$, estimate $E$, and neural response $R_{i,t}$ at time $t$ in principal component $i$ in the same trial. II quantifies the part of information in neural responses that is

common to both stimulus and estimation information. II is computed as the minimum between two terms with similar but slightly different interpretations, as follows:

$$II(S;R_{i,t};E) = \min[SI(E;\{S,R_{i,t}\}), SI(S;\{E,R_{i,t}\})] \quad (2)$$

The first term is $SI(E;\{S,R_{i,t}\})$, the information about estimation $E$ shared between stimulus $S$ and neural response $R_{i,t}$. This quantity is obtained (using the definition of shared information derived in ref. [59]) by computing the maximum shared dependency between $S$ and $R$ conditioned on $E$, defined as the mutual information between stimuli and neural responses minus the mutual information between stimuli and responses conditioned on choice, with the maximum computed over the space of all distributions q($s,r_{i,t},e$) that preserve the marginals p($s,e$) and p($r_{i,t},e$). The second term is $SI(S;\{E,R_{i,t}\})$, the information about $S$ shared between estimate $E$ and $R_{i,t}$. This term is obtained in a similar way by computing the maximum shared dependency between $E$ and $R_{i,t}$, conditioned on $S$. Minimizing between the two terms ensures that II satisfies key properties that would be expected from a measure with this interpretation, including that independent $S$ and $R_{i,t}$, that intersection information is non-negative and bounded by the stimulus and estimation report information encoded in neural activity, and by the information between stimulus and estimation report. Intersection information uses the so-called "redundancy" term of the Partial Information Decomposition and is thus particularly data robust (see Supplementary Fig. 12 and ref. [60]), as it uses combinations of bivariate probabilities of $S$ or $E$ vs neural response $R_{i,t}$, instead of the full trivariate probabilities, which are more difficult to sample. We again discretized $S, R_{i,t}$ and $E$ into 3 bins (see below), matching the number of trials when comparing II between conditions.

We chose 3 bins for all main analyses reported in this paper because this yielded a good number of observations per bin of the bivariate distributions used in the information theoretic calculations (and thus more accurate and unbiased information estimates[57,61]) for all analyses, including the bivariate probability distributions required for computing intersection information[57,58,62], even after splitting data by consistency. For the less sampled analyses (i.e., those focusing on the "near-zero trials", splitting by sample consistency, and pooling across two sample positions (e.g., 7 and 8), the median trial numbers per computed information measure were 207 (range: 84–263) for the Cue condition and 191 (range: 87–257) for the Choice condition, respectively. These trial counts yielded at least 9 observations per bin (median 22 observations) for all information measures reported (for both Cue and Choice conditions). For the computations of neural information measures across all trials (Fig. 3), the median number of observations per bin was 31, for both Cue and Choice.

We tested the stability and validity of the information measures presented in this paper in two ways. First, we verified empirically that the qualitative patterns (time courses and cortical distributions) of all information theoretic measures assessed here were stable over a range of bin numbers (3, 5, 7, 9 bins) for both task conditions (Supplementary Fig. 12a–f, compare with Fig. 3). Second, we performed simulations, in which we computed the information measured as function of the number of simulated trials and showed that the amount of data used in our study was comfortably sufficient to achieve a correct value of all the information quantities with the limited-sampling bias corrections used in this paper (Supplementary Fig. 12g–i).

To conservatively correct for the bias in information estimation due to limited sampling[57,61], we subtracted the average information after shuffling the respective task variable $X$ (i.e., $S$ or $E$) across experimental trials from all information-theoretic measures (see Supplementary Fig. 12g–i for a simulation of the effectiveness of this procedure). We used $N = 100$ shuffles for all the main analyses of consistency effects reported in this paper ($N = 20$ shuffles for overall

information measures in Fig. 3). To eliminate possible confounding effects of possible residual errors in information estimation (which depend on the number of available trials[57]) when comparing information values between consistent and inconsistent conditions, we matched the number of trials by random subsampling ($N = 5$) of the condition with more trials (typically consistent) to match the trial number of the other condition when computing information.

**Trial selection for consistency analyses.** Our design entailed systematic trial-to-trial variations in the generative mean, which were important for demonstrating stimulus-guided choice and estimation behavior and keeping participants engaged, but introduced correlations stimulus samples, categorical cue or choice, and estimation reports. In the ideal, the analysis of psychophysical kernels would focus only on trials with generative means of 0°. In practice, it was necessary to use a small range of generative means around 0°, so as to obtain a sufficiently large number of trials for the key analyses of consistency effects (small counts of inconsistent samples). The range of means from −4° to +4° was the smallest range that yielded sufficient numbers of observations necessary for the computation of II(S;R;E). We verified through simulations of the synthetic agent based on the above-described ideal that this range did not yield spurious consistency effects in I(S;E). Here, we simulated the agent simulated for (i) varying levels of "decision noise" (fractions of trials with random judgments) giving rise to varying levels of choice accuracy and (ii) varying subsets of trials with different ranges of generative means surrounding 0°. As for the actual participants, we then computed I(S;E) after sorting samples as consistent or inconsistent with respect to the intermittent choice. For noise levels matching the participants' accuracy for the range of means from −4° to +4° (Fig. 1c), the stimulations showed no difference between consistent and inconsistent conditions in in this "near-zero range". Such a difference would have been indicative of a spurious effect because the simulated agent was unbiased. Furthermore, when spurious consistency effects occurred for larger ranges of generative means and/or larger choice accuracies, those were present both in the second and in the first interval, unlike what we observed in the data (Fig. 2). We used the same subset of trials for all analyses of the effects of consistency on neural information measures entailing evidence samples and behavioral estimates (Figs. 4, 5). We restricted the computations of information measures to participants with at least 81 trials for each computation of the above probability distributions for each participant and condition. This led to the exclusion of up to three participants from the analyses of consistency effects on neural encoding and readout, yielding $N = 27$-$29$ participants ($N = 29$ for all main results).

**Regression analyses of behavior and neural activity.** We used linear regression to quantify the psychophysical kernels of each sample as a measure of contribution to the final estimation of the generative mean. For the behavioral data, we regressed the estimation ($E_{tr}$, $tr$: trial) with each sample ($s_{i,tr}$, $i$: sample position, $tr$: trial):

$$E_{tr} = \beta_i^0 + \beta_i^1 \cdot s_{i,tr} \quad (3)$$

The (signed) regression coefficients $\beta_i^1$ quantify the relationship between the final estimate and stimulus samples $s_{i,tr}$ at a given position $i$ in the sequence. To enable direct comparison with the temporal profiles of the strength of evidence weighting based on the (unsigned) mutual information reported in the main paper, we took the absolute values of those coefficients and compared them between consistent and inconsistent conditions: weaker impact of inconsistent samples on the estimate should translate into smaller magnitudes (i.e., absolute values) of regression coefficients for inconsistent compared to consistent samples.

For the neural data, we used a 10-fold cross-validated decoding approach. We fitted the following linear regression models for each PC $j$ and sample position $i$ at each time point $t$ in the training data:

$$s_{i,tr} = \beta^0_{j,i,t} + \beta^1_{j,i,t} \cdot PC_{j,t,tr} \qquad (4)$$

$$E_{tr} = \beta^0_{j,i,t} + \beta^1_{j,i,t} \cdot PC_{j,t,tr} \qquad (5)$$

We then quantified the stimulus decoding accuracy in the held-out (test) data, in terms of the correlation between the stimulus variable predicted by the decoder and the actual stimulus variable. We repeated this procedure for all ten folds and averaged the resulting correlations. We developed an analogous procedure for quantifying the readout of neural stimulus information (i.e., linear analog of II(S;R;E)). To this end, we computed the correlation between the neural decoder-predicted sample stimulus and the behavioral estimate of the participant, as an intuitive quantification of how well the information encoded in neural activity predicts the behavioral report. Each of the above analyses yielded several correlation coefficients, one per principal component, which we refer to as corr($\hat{s}$, E), with $\hat{s}$ denoting the stimulus sample predicted by neural stimulus decoder. These values were across principal components to obtain time courses of corr($\hat{s}$, E). To enable direct comparison with the (unsigned) information theoretic results presented in the main paper, we averaged corr($\hat{s}$, E) across the time interval of maximum stimulus information (same as for all figures) and again took absolute values.

We performed univariate rather than multivariate regression analyses (i.e., separately regressing individual stimulus sample values, or neural activities of individual principal components, on the respective target variable) to maximize the similarity to the main information-theoretic analyses described in the previous subsection.

**Statistical analyses.** Wherever possible, we used permutation tests (number of permutations: 2000, otherwise specified) to assess the statistical significance of differences between conditions (e.g., consistent versus inconsistent). The thresholds corresponded to $p = 0.01$ or higher when testing against baseline or $p = 0.05$ when testing between two conditions. For time courses, we used a cluster-based permutation test (clusters in the time domain). For statistical maps, we averaged information measures across the interval $0.1\,s - 0.5\,s$ from sample onset and performed one-sided permutation tests for each area group (average across hemispheres and parcels per area, see above), and used FDR correction to control for multiple comparisons.

For the comparisons between encoding and readout of sample information in cortical population activity, support for and against the null hypothesis were both of importance for the conclusions. Hence, we complemented the $p$-values from permutation tests shown in the corresponding bar graphs (insets in main Fig. 4 and associated figures in the Supplementary Information) with Bayes factors that quantify support for the null hypothesis. We calculated the Bayes factors (abbreviated as BF) based on the t-statistics obtained from two-sided paired $t$-tests[63,64]. BF values of between 0.3 and 0.1 provide strong (decisive for BF < 0.1) support for the null hypothesis[65].

We used 2-factorial ANOVAs to test for the interaction between the factors evidence strength and condition on overall estimation bias. Effect size was calculated as the partial eta-squared ($\eta p2$)[66] and 95% confidence intervals were computed based on the point estimate ($\eta p2 \pm t_{crit}*SE$).

We quantified and tested spatial pattern similarity by correlating (Pearson correlation coefficient) the across-area patterns of the neural consistency effects (consistent - inconsistent) obtained in different conditions (i.e., Choice and Cue) or in different measures of neural encoding or readout (i.e., I(S;R), I(R;E), II(S;R;E), or corr($\hat{s}$,E)). These pattern correlations were computed both at the level of the original

180 areas from the parcellation and at the level of the 22 area groups used for all other analyses (see section Parcellation and regions of interest (ROIs)). Patterns of cortical population activity tend to exhibit spatial autocorrelation, leading to inflation of statistical significance when assessed with standard approaches[67,68]. We used a rigorous permutation procedure for obtaining $p$-values for pattern correlations that accounted for the spatial autocorrelation structure[67,68]. To this end, we generated null models using a spin rotation method which projects the cortical surface onto a sphere and then randomly rotates, generating permuted cortical surfaces with preserved spatial auto-correlations. We used the spherical projection of the FreeSurfer fsaverage surface for the atlas (https://github.com/rudyvdbrink/Surface_projection) to assign each parcel with the coordinates of the vertex that was closest to the center of mass of the parcel. We then reassigned each original parcel with the value of the closest rotated parcels. We repeated this procedure 10000 times and compared the original correlation coefficients against the so-obtained null distribution (of 10000 correlation coefficients between the null models and the original data), yielding $p$-values.

**Information-theoretic and linear analyses of simulated behavior and neural activity.** To test information theoretic and regression-based analyses and illustrate possible neural mechanisms underlying selective evidence weighting, we developed a simple model of encoding of sample information and readout to generate an estimate based on the activity of a number of neurons (or neural populations) in the brain. Here, we take the activities of neural populations to be analogous to the activities of the different principal components extracted from the MEG data for each region of interest. For simplicity of visualization, we present results for two neuronal populations, but we verified them to hold regardless of the number of simulated neural populations. The model generated Gaussian neural activity for each neural population which was then turned into a sample and an estimate exactly with the encoding and decoding regression models used to fit the data, which contained separate regression weights for each neural population, and which were corrupted by separate sources of Gaussian noise for encoding and decoding (Eqs. 4,5). Since sums of Gaussians are Gaussian, this procedure was formally equivalent to first generating Gaussian samples and transforming those linearly into neural activity and then behavioral estimates as illustrated in the schematic of Supplementary Fig. 5a. We chose the implementation described above for two reasons: (i) to match the stimulations directly with our data analyses (Eqs. 4,5), and (ii) to enable fixing the level of information about samples or estimates while rotating regression vectors. The parameters setting the level of encoding and decoding noise could be varied separately across simulations, to study their respective effect on selective evidence weighting. The encoding and decoding regression coefficient could also be separately varied across simulations. Matched relative weights between encoding and readout correspond to efficient readout of the encoded information[17]. Mismatched encoding and decoding weights correspond to inefficient readout. In our data, consistent (inconsistent, respectively) samples have on average the same (opposite, respectively) sign of the final estimate (Supplementary Fig. 4a, b). This is because consistent samples tended to have the same sign of the generative mean, and the generative mean tended to have the same sign of the final estimate, and conversely for inconsistent samples. Thus, we simulated the transformation of consistent (inconsistent) samples using encoding weights of a sign equal (opposite) to the sign of the readout weight. This created samples and estimates with, on average, equal (opposite) signs. Results of the simulations are presented in the different rows of Supplementary Fig. 5 for five different parameter settings: one case of consistent samples with efficient readout (Supplementary Fig. 5b); one case of inconsistent samples and similarly efficient readout (Supplementary Fig. 5c); and three cases of inconsistent samples, all of which yielded the reduction of impact of stimulus samples on estimates as we observed in

our behavioral data (Supplementary Fig. 5d–f; see leftmost scatter plots and I(S;E) values). These three cases entailed different mechanisms, which operationalized the neural hypotheses, between which our study aimed to arbitrate: reduced encoding precision (Supplementary Fig. 5d) versus impaired readout (Supplementary Fig. 5e, f); the latter through increased decoding noise (Supplementary Fig. 5e) or mismatch between encoding and readout (Supplementary Fig. 5f). The model results shown in Supplementary Fig. 5 were obtained by applying the same information theoretic and linear regression analyses described in the previous sections for the empirical behavioral neural data to the simulate data.

### Reporting summary

Further information on research design is available in the Nature Portfolio Reporting Summary linked to this article.

## Data availability

The behavioral data and preprocessed neural data (as time courses of principal components, information measures, and linear correlations per cortical region) including source data for the main figures and the figures from the Supplementary Information data generated in this study have been deposited in a persistent repository of the University of Hamburg Center for Sustainable Research Data Management under accession code https://doi.org/10.25592/uhhfdm.16918. The raw MEG and MRI data are protected and are not publicly available due to data privacy laws. Raw MEG data will be shared upon request addressed to: t.donner@uke.de.

## Code availability

All custom code to reproduce the reported results and figures is available on Github: https://github.com/DonnerLab/2025_Park_ConfirmationBias-through-Selective-Readout-of-Information-Encoded-in-Human-Parietal-Cortex. https://doi.org/10.5281/zenodo.15350393.

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

## Acknowledgements

We thank Maxime Maheu, Joshua Calder-Travis, Ruud van den Brink, Anke Braun, Gina Monov, Marlene Petersson, and Emma Krink for help with data collection. This work was funded by the Deutsche Forschungsgemeinschaft (DFG, German Research Foundation) project DO1240_2-2 (to T.H.D.), the German Federal Ministry of Education and Research (BMBF) project 01GQ1907 (to T.H.D.), the National Science Foundation (NSF), project IIS-1912232 (to A.A.S.), and by the NIH Brain Initiative, grant R01NS108410 (S.P.).

## Author contributions

Conceptualization: B.C.T., A.A.S., S.P., T.H.D.; methodology, software, investigation: H.P., A.A., B.C.T., M.C., S.P., A.A.S., T.H.D.; data curation: H.P., A.A., B.C.T.; formal analysis, visualization: H.P.; writing (first draft): H.P., T.H.D.; writing (review and editing): H.P., A.A., B.C.T., M.C., S.P., A.A.S., T.H.D.; supervision: S.P., A.A.S., T.H.D.; funding acquisition: S.P., A.A.S., T.H.D.; project management: T.H.D.

## Funding

## Competing interests

The authors declare no competing interests.
