## [Transparent Peer Review file · Nature Communications]

Confirmation Bias through Selective Readout of Information Encoded in Human Parietal Cortex

Corresponding Author: Professor Tobias Donner

Version 0:

Reviewer comments:

Reviewer #1

(Remarks to the Author)

The manuscript asks whether choice biases confirming previous choices in perceptual decisions arise from biased percepts or biased action selection upon unbiased percepts. It combines human perceptual orientation estimation experiments with analysis of MEG data to address this question. Humans are asked to judge the average orientation of a sequence of 12 oriented patterns. After 6 oriented patterns they are furthermore either asked to perform an interim choice, or are cued about whether the average orientation is clockwise or counterclockwise from vertical. In both cases, choices/cues biases how subsequent orientation percepts contribute to the decision makers' final choices. MEG data suggests that these biases do not arise from a biased processing of the individual orientations, but instead from biases in how unbiased percepts are turned into orientation estimates.

The studies sheds new light on the origin of confirmation biases in estimation and decision making. It does so in a concise way with an elegant experimental paradigm. Unfortunately, it only provides limited analysis of the resulting behavior, and relies on fairly indirect analyses to support their main claims. In particular,

- The authors identify a different impact of interim choices vs. cues (e.g., Fig. 1f vs 1g). However, these differences are not reflected in the final estimates (Fig. 1d grey vs. black). Is there indeed no difference in these final estimates, or would there be an effect if these estimates are conditional on interim choices/cues? If there is really no difference, how can this be brought in line with Figs. 1f-h?

- The analysis focuses on biased vs. unbiased percepts of individual orientations, and how they drive reported orientation estimates. They authors find that orientations presented directly after interim choices/cues contribute in the most biased way to the final estimates. They furthermore claim that the percepts themselves are not biased, but that it is instead how they contribute to the final estimates that introduces these biases. However, they do not elaborate how this would work in terms of the algorithms that the brain uses to form these estimates. In particular, given that evidence is accumulated across orientation samples, how could the orientation samples be perceived without bias, but contribute in biased ways to the final estimate? Shouldn't this be visible in the process that performs this evidence accumulation? Previous studies have queried this process from MEG data, such that it should be possible to perform similar analyses with the data available to the authors. Having some estimate of biases in the evidence accumulation process would significantly strengthen the author's conclusions. More generally, it would be really helpful to have a model that, at least qualitatively, describes how the authors imagine these biases to arise exactly. At the moment, this is hard to understand from the provided analyses and text.

- Most of the analyses rely on mutual information and intersection information to assess the relation between behavioral and neural measures. These information measures are hard to estimate, might be biased, and require coarse binning of the data to estimate the required joint probabilities. For neural data in particular, the manuscript only provided an average of these information measures across individual estimates for different principal components of the original data. While the authors made sure to minimize the impact of estimation biases, and confirm that binning has limited impact on their final conclusions, it would have been welcome if the authors would have used more direct measures to assess these relationships. For example, wouldn't it have been possible to use linear regression or similar to assess the impact each orientation percept has

on the final estimates (as shown in Fig. 1f/g)? The same applies to analysis of the neural data in Fig. 2b-e. For these purposes, having a model (or a set of competing models) for how the percepts are being processed would again really help.

Overall, I think that further analyses would be required to make the authors' conclusions convincing. Having a model describing how biases arise in the evidence accumulation process would probably aid these analyses.

Minor details:

Fig. 1c: why isn't the accuracy in the cue condition at 75%. Wasn't the cue correct 75% of the time. If the deviation from 75% arises due to focusing on near-zero orientations only, why is it different from these orientations?

Fig. 3: readers would benefit from an elaboration why $I(S;R)$ and $I(R;E)$ can show no significant differences between consistent and inconsistent stimuli, but $I(S;R;E)$ can. This seems counterintuitive at first.

l491: "psychometric function logistic function" - repetition

l592, Eq. (2) - seems to be missing closing curly brackets

Fig. S1 a/b: same scaling of horizontal axis would aid comparison between a and b

Reviewer #2

(Remarks to the Author)

This study examines the effect of choice-induced biases in evidence weighting ('confirmation biases'), with a focus on whether these biases materialise at the level of initial encoding of sensory information, or at the transformation into a subsequent behavioural response. The study finds support of the latter hypothesis. The authors use a clever manipulation to compare self-generated reports ('choice-induced' biases) vs externally cued biases ('cue-induced' biases). Although both have an influence on subjects' behaviour, the former has a stronger influence, and this is also seen at the level of neural analyses.

This is a solid paper that builds well upon recent findings in the field of sensory evidence integration, and how information that is (in)consistent with the participant's current belief is encoded and affects subjects' behaviour. The behavioural and neural analyses are mostly very convincing and the conclusions appear robust and reliable. I had a couple of questions about the analysis strategy adopted, one behavioural and one neural, that I hoped the authors could address in a revision.

1. Behavioural analysis: For the $I(S;E)$ analyses, the authors focus exclusively on trials that have a generative mean close to 0, pointing out in the methods section that the ideal situation for their analyses would exclusively include trials with a true mean of 0. I couldn't quite see whether another analysis might not allow the authors to include all trials in their analyses? For example, Cheadle et al (Neuron, 2014) characterised psychophysical kernels with logistic regression to characterise the impact of each sample on the eventual choice of participants. Wouldn't a similar approach be possible here, using linear regression to predict the subjects' orientation response at the end of the trial? Would this allow the authors to make use of all trials rather than just a subset of trials, and not require binning of responses (as they allude to in the methods of the current approach)? I can see the appeal in using the same approach as they used to characterise the neural responses, but it did seem to have potential downsides in discarding so many trials.

2. Dimensionality reduction: The top of figure 2a implies that after reducing dimensionality within a region of interest, and computing $I(S;R;E)$, the authors took the approach of simply averaging across all the M principal components to reconstruct the timecourse. Why is averaging the appropriate thing to do here? It seems like it might be equally helpful just to focus on the first principal component, or to weight the principal components by the amount of variance that they explain across the N vertices, especially given that $M=1$ for some brain regions, but $M=15$ for others. Does this step have a strong impact upon the results?

3. There appears to be a mistake in figure 4 legend – (a) is said to refer to $I(S;E)$, whereas (b) is said to refer to $I(S;R;E)$, while the reverse appears to be true on the figure.

Reviewer #3

(Remarks to the Author)

This manuscript investigates the origin of confirmation bias in decisions based on summarizing a sequence of stimuli. In the middle of this sequence, participants were either asked to make a two-alternative forced choice on the previous samples or a pre-assigned button press (based on a 75% valid cue). Using information theoretic measures, the authors show that — behaviorally — sensory evidence after the choice is upweighted when consistent with the choice. Using MEG source reconstruction, they then show that another measure ('intersection information') i) differs between congruent and incongruent sensory evidence after the choice and ii) correlates with individual behavioral bias. The authors conclude on a decisional rather than a sensory origin for the confirmation bias.

The experimental paradigm is elegant, as is the use of state-of-the-art MEG source reconstruction. However, I found the

conclusion a bit incremental compared to previous work from the same group. The task is sold as a 'novel behavioral task' (line 65) but it seems very similar to Talluri, Urai et al, 2018 (Current Biology) for the non-specialist. I understand that the main claim is that the confirmation bias does not affect the sensory encoding itself but how much sensory information in the neural response affects the decision. However, in the 2018 paper, the confirmation bias was also found in a mathematical task in which (unless I am missing something) numbers are not sensed differently according to previous choices. I think the authors should clarify how viable is the alternative explanation that is falsified by this study.

The measure on which the paper's main claims are based ('intersection information') is hard to grasp. Conceptually, it was unclear how the interpretation that it quantifies 'the amount of stimulus information encoded about the stimulus S by the neural signal R that informed the behavioral estimation report E' (lines 584-7). As with any complex method, it would be reassuring to see the underlying data (source-reconstructed MEG activity). But here, one can only blindly trust that the method does its job. Methodologically, it seems that this 'intersection information' is computed with as low as 4 trials per bins and is normalized by a baseline obtained by only 20 permutations. One could suspect that this is a very unstable measure.

Neural measures are also computed in a task with fast presentations of stimuli (1 sample every 150 ms?) which I assume induces overlapping neural responses. In Figure 3, it seems that the effect lasts between about 0.1 to 0.5s post-stimulus onset and therefore the measures are also affected by the pre-choice activity generated by samples 5 and 6 that are not necessarily independent from samples 7 and 8 because of the subsampling of consistent and inconsistent samples, right? Considering the complexity of the measures I could not convince myself that the data support the conclusions. This concern applies much less to the analyses done on behavior only, for which the authors used additional simulations (lines 608 - 634) to test for some possible confounds. A similar analysis with the neural data would have helped convince.

Minor

- Why were the conditions cue/choice not randomized and counterbalanced?
- Line 600: is it not rather the '*estimation* E' than the 'stimulus E'? This paragraph is specially hard to follow.

Version 1:

Reviewer comments:

Reviewer #1

(Remarks to the Author)

The authors have significantly strengthened their manuscript by refining existing, and adding new analyses, and by improving clarity of their writing. I appreciate them de-emphasizing the difference between the interim cue and choice condition, as this was indeed confusing my reading of the previous version of the manuscript. Furthermore, supporting all information-based measures with measures based on linear regressions removes any doubt about whether these information-based measures were biased. Their simulations in the new Fig. S5 also provided some intuition into their results that was lacking in the previous version of the manuscript.

Overall, I have no further comments on the manuscript.

Minor:

Figure 4 legend: "See main text for and simulations in Figure S5 for details." - first "for" should be removed.

(Remarks on code availability)

Reviewer #2

(Remarks to the Author)

The authors have appropriately addressed all my comments from the first round of review, and have provided a particularly thorough response on the relationship between information theory and regression-based approaches to analysis of the data.

(Remarks on code availability)

I reviewed the code by examining the GitHub repository online, but I did not attempt to install and run the code myself. The code includes a README file, with a functioning link to the University of Hamburg repository where the data is stored, and clear links to toolboxes on which the analysis depends (e.g. for Information Theory-based analyses). The code is clearly laid out, in that the MATLAB scripts are named so that Figure X is created by FigureX_YYY.m - this will help readers who are interested in recreating a particular analysis to straightforwardly reproduce the reported figures.

Reviewer #3

(Remarks to the Author)

The authors have answered all my concerns with convincing explanations, additional data/analyses and textual revisions.

I now think it is a great paper.

Thank you and congratulations.

(Remarks on code availability)

There is a README file and there are matlab scripts for every figure in the paper. I did not download the data to test the code (> 200 Gb).

NCOMMS-24-43181-A: Response to reviewers

We thank the reviewers for their thoughtful and constructive evaluation of our manuscript. To address the reviewers' concerns, we have performed a substantial number of new analyses and modeling culminating in multiple new figures. We have also implemented major textual revisions throughout all sections of the manuscript, which are highlighted in red in the text. Taken together, these revisions have substantially strengthened our paper and provided additional support for our conclusions.

In what follows, we provide a detailed point-by-point reply to each of the reviewers' comments (printed in *blue italics*). Some reviewer points contained multiple questions/ comments. We have opted to reply to each individual question/comment in turn.

We hope that you will now find our manuscript suitable for publication in *Nature Communications*.

Sincerely, on behalf of all authors,

Hame Park & Tobias H. Donner

Reviewer #1

R1.1. The authors identify a different impact of interim choices vs. cues (e.g., Fig. 1f vs 1g). However, these differences are not reflected in the final estimates (Fig. 1d grey vs. black). Is there indeed no difference in these final estimates, or would there be an effect if these estimates are conditional on interim choices/cues?

We followed your suggestion to analyze the mean estimates conditioned on the interim cue or choice (Figure S2a,b). The corresponding mean estimates conditioned on choice/cue category are now shown in the new panel Figure 1h, which exhibit a clear difference between estimates following right or left category "events", regardless of whether that event is a choice or a cue.

In Figure 1i, we now also show the difference in mean estimates for 'right' versus 'left' categories as function of evidence strength. This difference increases as function of category evidence strength for the Choice condition, but not the Cue condition. A simple ideal observer model (noise free and unbiased, ignoring the Cue) does not exhibit this pattern. This pattern can be explained by the notion that choice-induced biases depend on the confidence in the associated event (which increases with evidence strength for a choice, but not for a cue of known reliability).

We highlight that we identified a significant impact of both, interim choices and interim cues, on the evidence weighting in these figures (formerly Figure 1f-h; moved to Figure 2 in the revised manuscript). The tendency of this effect being larger for Choice than for Cue is a second-order finding, and not the central point of our paper.

Please note that also the neural effects of consistency on intersection information exhibit similar effects for Cue and Choice conditions (Figure 4). Indeed, after re-running our analysis of intersection

information with 100 shuffles for correction (in response to a comment from Reviewer #3; previously: 20 shuffles), the consistency effect on intersection information in the parietal region of interest is statistically significant even after correction for multiple comparisons across all brain regions (previously only without correction). To further highlight the similarity of the consistency effects on neural intersection information, we have now expanded Figure 4 by an analysis of the similarity of the cortical maps of consistency effects. We find clear and significant similarity (spatial pattern correlation) of the consistency effects on regional intersection information. As negative control, we show that there is no such correlation for the consistency effects on stimulus information (and estimation information), which show no significant effects anywhere.

In sum, there is a clear correspondence of consistency effects at the level of behavior and the brain for both the Choice and Cue contexts. We apologize if our tentative interpretation of the difference between the psychophysical kernel effects in Choice and Cue in the previous version of our manuscript gave the misleading impression that this difference between conditions is the key finding. The Cue condition simply serves the role of a useful reference condition, with which to compare any confirmation bias effects in the Choice condition, because the Cue had a known and similar validity to participants' choices. We have clarified this by textual revisions throughout the Results and Discussion sections, toning down any statements pertaining to the difference between the consistency effects in Choice and Cue conditions.

Your comment also made us realize that the mean estimates shown in the previous version of Figure 1 (former panel d) did not do a good job in capturing participants' estimation behavior, including any biases therein. We have now replaced those by histograms of the estimation reports (group level) as function of the evidence source in Figure 1e,f, and collapsed across generative means for both conditions, along with formal tests of unimodality (Figure 1g). We also show the underlying individual estimation distributions, again with tests of unimodality per condition, separately for each participant (new Figure S1). Indeed, those appear similar between interim choices versus cues in for most participants. These new analyses provide clear evidence for repulsive biases in participants' estimates, in both Cue and Choice conditions, which may originate from several sources including (but not limited to) the interim cues/choices. This is now unpacked in the corresponding Results text.

If there is really no difference, how can this be brought in line with Figs. 1f-h?

These psychophysical kernels (formerly Figure 1f-h; now moved to Figure 2 in the revised manuscript) show a difference in the association of sample fluctuations and behavioral estimates. They show clear behavioral effects of consistency for both Cue and Choice, whereby this effect is slightly, but significantly larger for Choice. These consistency effects for both conditions, and therefore also their condition difference, are local in time, restricted to two samples in a sequence made up of twelve samples. This temporal specificity can explain the difference between effect strengths for Cue and Choice in the measures shown in Figure 2 (psychophysical kernels) and Figure 1 (estimate distributions).

Indeed, this issue highlights why the high temporal precision of the psychophysical kernel analysis at the levels of the processing of individual samples is so important in isolating the effect of interest. We now make this more explicit in the corresponding part of Results, thereby motivating our kernel analysis. We also elaborate on this point in a dedicated paragraph of Discussion (p.13, 3rd par).

Again, we have also toned down any statements regarding the difference between the consistency effects in Choice and Cue throughout the paper – which, we consider a subtle quantitative, not a qualitative difference.

Finally, we point out that effects other than weighting of the evidence (captured by the psychophysical kernels) may also contribute to a difference in mean estimates in the analysis you suggested (new Figure 1h). For example, we have shown in our previous behavioral work using a similar task design (Talluri et al, 2018), that an additive shift of an internal decision variable induced by choice or cue can produce a right versus left difference in mean estimates, but may not be evident in an analysis of evidence weighting.

R1.2. The analysis focuses on biased vs. unbiased percepts of individual orientations, and how they drive reported orientation estimates. They authors find that orientations presented directly after interim choices/cues contribute in the most biased way to the final estimates. They furthermore claim that the percepts themselves are not biased, but that it is instead how they contribute to the final estimates that introduces these biases. However, they do not elaborate how this would work in terms of the algorithms that the brain uses to form these estimates.

We have now performed simulations of a simple neural encoding and decoding model to shed light on how neural computations may differ between consistent and inconsistent evidence samples. The model simulations are shown in the new Figure S5. We use the model to understand how each of the measures (information-theoretic and the new linear regressions introduced in revision; see below) that we apply to our behavioral and neural information measures are affected by different computational mechanisms that may give rise to biased contributions of evidence to estimates. For simplicity, we restrict the model to the transformation of a single evidence sample into a single estimate, but the insights translate without loss of generality to contexts that entail the integration of multiple sequentially presented samples. In the model, stimulus samples are encoded linearly in the strength of neural activity in two neural populations, and the resulting neural activity pattern is then decoded to produce an estimate, by applying linear weights to the activity of each population. Encoding and decoding are both corrupted by separate sources of noise.

In Figure S5, we consider three possible types of changes in neural activity with inconsistent samples. In the first case, the encoding noise increases. In the second and third cases, the readout deteriorates, in two different ways: decoding noise increases, or (without any change in encoding or decoding noise), the encoding and readout weights become misaligned. We show that each of these three cases can give rise to the observed effect in the psychophysical kernels (less contribution of inconsistent samples to estimate). But only the third case (misaligned readout) produces the pattern of neural measures we observe in the data – and this is true both, for the information theoretic measures from our original paper and the new linear regression measures we have added in revision (see below).

Thank you for pushing us on this point: We feel that these simulation results helped to solidify the intuitions about candidate mechanisms at play and therefore strengthen our conclusions.

In particular, given that evidence is accumulated across orientation samples, how could the orientation samples be perceived without bias, but contribute in biased ways to the final estimate? Shouldn't this be visible in the process that performs this evidence accumulation? Previous studies have queried this process from MEG data, such that it should be possible to perform similar analyses with the data available to the authors. Having some estimate of biases in the evidence accumulation process would significantly strengthen the author's conclusions.

The idea is conveyed by the new simulation from Figure S5 (bottom row) and schematic of Figure 4k: The encoding and readout have similar strength between consistent and inconsistent samples, but encoding and readout weights are mismatched when samples are inconsistent. As a consequence, the stimulus information encoded in parietal cortex contributes less to the behavioral estimate.

We agree that it will be interesting to further dissect the “readout” operation in terms of the different transformations that occur downstream from the encoding of sample information in our task – in particular, the sequential accumulation of the evidence samples. A change in readout weights may be brought about by a dynamic up- or down-weighting of the sample representation in the accumulation process, similar to what we and others have found for the effects of computational variables other than consistency and for tasks requiring categorical choices (Murphy et al., 2021). Led by the Stocker lab, we are currently developing a resource-rational belief updating model for our current continuous decision task that captures multiple types of biases in shaping evidence accumulation profiles (including choice-/cue-induced and temporal biases) in terms of performance-effort tradeoffs. This model is novel and developing it is challenging, so we consider this a full study in and of its own. Likewise, we are not aware of previous neurophysiology work that has delineated neural decision variables (in terms of integrated evidence) for continuous decisions in primate cortex. This is beyond the scope of this study.

The focus of our current study is on the stage of evidence encoding in the brain. This is conceivably the earliest level, at which neural processing may differ between choice-consistent and choice-inconsistent evidence – as we here demonstrate. The goal of our study was to arbitrate between different mechanisms, by which such a difference at the encoding stage may come about. We have now clarified our focus and goal in the Introduction. Our study presents comprehensive psychophysical and neurophysiological quantification of behavior and neural activity tailored to this goal, and a clear pattern of results.

R1.3. More generally, it would be really helpful to have a model that, at least qualitatively, describes how the authors imagine these biases to arise exactly. At the moment, this is hard to understand from the provided analyses and text.

We hope that our revisions summarized under point R1.2 help clarify the reasoning: the model results from new Figure S5 illustrate how these biases can arise, and which of the alternative mechanisms are most in line with our data.

R1.4. Most of the analyses rely on mutual information and intersection information to assess the relation between behavioral and neural measures. These information measures are hard to estimate, might be biased, and require coarse binning of the data to estimate the required joint probabilities. For neural data in particular, the manuscript only provided an average of these information measures across

individual estimates for different principal components of the original data. While the authors made sure to minimize the impact of estimation biases, and confirm that binning has limited impact on their final conclusions, it would have been welcome if the authors would have used more direct measures to assess these relationships. For example, wouldn't it have been possible to use linear regression or similar to assess the impact each orientation percept has on the final estimates (as shown in Fig. 1f/g)?

We now include a version of the psychophysical kernels based on linear regressions in Figure S4. These show effects very similar to the information measures in the new main Figure 2.

The same applies to analysis of the neural data in Fig. 2b-e. For these purposes, having a model (or a set of competing models) for how the percepts are being processed would again really help.

We have also added corresponding linear regression versions of the neural information measures, including a newly developed linear analogue to intersection information (Figure S11). These results also match those of the neural information measures, in terms of time courses of effect within our parietal region of interest as well as in terms of the spatial similarity of the cortical maps of effects.

We would like to highlight that linear regression methods to derive the impact of information encoded in neural activity to behavioral estimates did not exist in the literature. We derived them with a methodological advance associated with our additional work. Previous studies have estimated psychophysical kernels based on linear regressions of stimulus samples on behavior, and/or used linear regression for neural encoding/decoding analyses. However, we are not aware of any linear analogues of the amount of sample information that is read out (the equivalent of intersection information) in the existing literature. We here now developed such an analogue for this revision, used simulations to rigorously test and compared it with the information-theoretic measures in the (ground-truth) simulated data, and finally applied to our actual neural data. This revealed effects of consistency on neural readout very similar to the main information-theoretic results we had already reported in the original submission. We think this novel methodology added to the paper is likely to be useful more broadly in the field, beyond the specific question consistency effects on neural readout that we address here.

That said, we prefer keeping all information measures in the main paper and presenting all regression analyses in the Supplement. This is for two reasons. First, information theory captures all associations between variables, linear or non-linear. Thus, it is better suited as tool to either discover or rule out, the presence of information or effects about information processing. For example, a lack of sample encoding detected with linear regression in a certain brain region may be either because that region encodes no sample information or because it encodes it non-linearly. Only the information theoretic quantification would allow to decisively conclude that there is no information encoding. Information theory also quantifies these interactions in a meaningful and well-defined scale (bits) which quantify reduction of uncertainty. Second, the measure 'intersection information' provides a direct and well-characterized measure of readout of neural information (Pica et al., 2017) that was not currently available with linear methods prior to our revision work. Therefore, we prefer to use the linear analogues we have developed here as secondary analyses to explain and understand simply the computations that we found to be significant with information theory.

Overall, I think that further analyses would be required to make the authors' conclusions convincing.

Having a model describing how biases arise in the evidence accumulation process would probably aid these analyses.

We hope that our new model simulations, analyses, and changes in the text help address your general concern.

Minor details:

Fig. 1c: why isn't the accuracy in the cue condition at 75%. Wasn't the cue correct 75% of the time. If the deviation from 75% arises due to focusing on near-zero orientations only, why is it different from these orientations?

For the trials with generative mean of 0° , both the cue category and the "correct category" were randomly selected. So, for those trials, the accuracy of both the cue and the choice are expected to be at 50% - which is what we find in the data (Figure 1d, the gray data point for $x=0$ is covered by the corresponding black data point at $y=0.5$). For all other trials the cue category matches the correct category with a probability of 0.75 (Figure 1d, gray data points. Please note that due to a small portion of trials discarded based on missed responses or bad MEG signals, the accuracy can be slightly different from exactly 75% in the Cue condition, hence the slight jitter in Figure 1d, gray data points). Thus, cue accuracy for all "near-zero" trials (means from -4 to $+4^\circ$) combined is between 50% and 75%. The choice accuracy, by contrast, remains $\sim 50\%$ for generative means slightly different from 0° (Figure 1d, black points). Hence, the choice accuracy for all "near-zero" trials combined is close to 50%.

Fig. 3: readers would benefit from an elaboration why I(S;R) and I(R;E) can show no significant differences between consistent and inconsistent stimuli, but I(S; R; E) can. This seems counterintuitive at first.

We hope that the conceptual geometric schematic now included in the new version of Figure 4, the simulation results from Figure S5, and the new linear decoding analogue of intersection information (Figure S11), all help to clarify this: The idea is that changes in the match between activity patterns related to sensory and behavioral report in a given brain region can, in principle, produce changes in intersection information without any concomitant change in the amount of information carried by the neural activity pattern about the stimulus or the behavioral report.

1491: "psychometric function logistic function" – repetition

Thank you. Fixed.

1592, Eq. (2) - seems to be missing closing curly brackets

Thank you. Fixed.

Fig. S1a/b: same scaling of horizontal axis would aid comparison between a and b

Thank you. We have now replaced these histograms by z-scored versions and plot both on the same x-axis.

Reviewer #2

R2.1. Behavioural analysis: For the I(S;E) analyses, the authors focus exclusively on trials that have a generative mean close to 0, pointing out in the methods section that the ideal situation for their analyses would exclusively include trials with a true mean of 0. I couldn't quite see whether another analysis might not allow the authors to include all trials in their analyses. For example, Cheadle et al (Cheadle et al., 2014) characterised psychophysical kernels with logistic regression to characterise the impact of each sample on the eventual choice of participants. Wouldn't a similar approach be possible here, using linear regression to predict the subjects' orientation response at the end of the trial?

We now include a version of the psychophysical kernels based on linear regressions in Figure S4. We use absolute values of the regression coefficients, because those capture the strength of the (linear) relationship independent of its sign. This is analogous to the information measures, which are always positive because they capture any way in which information is expressed, regardless of its sign. Indeed, the regression analyses show effects very similar to the information measures.

For completeness, we also now show linear regression analogues of the neural information measures, including a newly developed analogue of the key measure 'intersection information'. This analogue did not yet exist in the literature, so we developed it for the purpose of this revision and tested and validated it in our new simulations presented in Figure S5, and then applied to the neural data.

That said, we prefer keeping all information measures in the main paper and presenting all regression analyses in the Supplement. This is for two reasons. First, information theory captures all associations between variables, linear or non-linear. Thus, it is better suited as tool to discover, or rule out, the presence of information or effects about information processing. For example, a lack of sample encoding in a certain brain region measured with linear methods may be either because that region encodes no sample information or because it encodes it non-linearly. Only the information theoretic quantification would allow to conclude that there is no information. Information theory also quantifies these interactions in a meaningful and well-defined scale (bits) which quantify reduction of uncertainty. Second, the measure 'intersection information' provides a direct and well-characterized measure of readout of neural information (Pica et al., 2017) that was not currently available with linear methods. Therefore, we prefer to use the linear analogues we have developed here as secondary analyses to explain and understand simply the computations that we found to be significant with information theory.

Would this allow the authors to make use of all trials rather than just a subset of trials, and not require binning of responses (as they allude to in the methods of the current approach)?

Simulations show spurious effects also for the first interval in the regression approaches. In general, our simulations of relationships between evidence fluctuations and behavioral reports showed that both the information and the regression analyses could produce spurious consistency effects resulting from the correlation structure of the task (all samples in each trial drawn from the same generative distributions with a fixed mean) when performed on all trials. Those spurious effects are then evident already in the first stimulus interval. Focusing on the near-zero range of generative means is conservative and helps pinpoint the genuine effects of choice/cue on subsequent neural processing and evidence weighting.

We do note, however, that we use all generative means for all neural data analyses that are not split by consistency, specifically: Figures 1, 3 and Supplemental Figures S3, S11, panels a & b. The results verify that our data are overall lawful and as expected.

I can see the appeal in using the same approach as they used to characterise the neural responses, but it did seem to have potential downsides in discarding so many trials.

We understand the concern, but still maintain that it is vital to focus on the near-zero trials for all analyses of consistency, regardless of whether those use information theoretic measures or linear measures. Modeling work shows that large deviations from zero-evidence can produce biases in measures relating variations in evidence or neural activity to behavior, for both information theoretic and linear measures (Chicharro et al., 2021). Those statistical biases, in turn, may affect comparisons between consistent and inconsistent conditions in ways that would be difficult to pinpoint. We would therefore feel very uncomfortable with presenting versions of our main analyses that may be subject to such confounds. Our approach for quantifying psychophysical kernels and neural readout effects focusing on the near-zero trials is conservative and rigorous.

Please note that our conservative approach is inherited from reverse correlation approaches in psychophysics (e.g., (Kiani et al., 2008; Neri & Heeger, 2002) as well as the seminal approach used by Newsome and colleagues for characterizing ‘choice probabilities’ (CPs) in monkey cortex: there, monkeys performed visual motion discriminations across a wide range of motion coherences including 0%; the CP analyses focused entirely on the subset of ambiguous trials (0% motion coherence; (Britten et al., 1996).

We also do not think that discarding the “strong-evidence” trials has any substantial downside: We have specifically designed our study to yield a sufficient number of trials even when applying our conservative and selective approach for these most specific comparisons between consistent and inconsistent samples. To this end, we asked each participant to complete four long MEG sessions, yielding a total of 1.856 trials per individual.

To further strengthen this point, we now include a simulation in order to demonstrate that the amount of data used in our information measure computations. We computed the information metrics as a function of the number of simulated trials, and show our numbers are sufficient to achieve a good and stable value of the information quantities (Figure S12g-i).

R2.2. Dimensionality reduction: The top of figure 2a implies that after reducing dimensionality within a region of interest, and computing $\Pi(S;R;E)$, the authors took the approach of simply averaging across all the M principal components to reconstruct the timecourse. Why is averaging the appropriate thing to do here? It seems like it might be equally helpful just to focus on the first principal component, or to weight the principal components by the amount of variance that they explain across the N vertices, especially given that $M=1$ for some brain regions, but $M=15$ for others. Does this step have a strong impact upon the results?

Using the first set of principal components rather than only the first one is in line with common approaches in animal physiology (e.g., (Kiani et al., 2014; Mante et al., 2013; Peixoto et al., 2021)).

There is no guarantee that the first component would contain most, or even any, of the task-related information – it could largely reflect spontaneous, task-unrelated fluctuations in activity (e.g. global “noise”), which tend to be large in neural population recordings (Leopold, 2003). For the same reason, weighting the principal components by their eigenvalues may also not be optimal, since a substantial part of the variance captured by the eigenvalues may be task-unrelated.

That said, we now show that the pattern of consistency effects in the information measures is also present for the first principal component of dorsal visual cortex activity alone and similar to the pattern obtained for the average across components, with a clear effect of consistency on intersection information (sample readout) and no effect on stimulus sample encoding (Figure S8).

In sum, we think that our policy of averaging across components which explain 90% of variance across the vertices of source-reconstructed MEG activity is simple, unbiased, principled, and well in line with common procedures in the literature.

R2.3. There appears to be a mistake in figure 4 legend – (a) is said to refer to $I(S;E)$, whereas (b) is said to refer to $\Pi(S;R;E)$, while the reverse appears to be true on the figure.

Thank you. Fixed.

Reviewer #3

R3.1. The experimental paradigm is elegant, as is the use of state-of-the-art MEG source reconstruction. However, I found the conclusion a bit incremental compared to previous work from the same group. The task is sold as a 'novel behavioral task' (line 65) but it seems very similar to Talluri, Urai et al, 2018 (Current Biology) for the non-specialist. I understand that the main claim is that the confirmation bias does not affect the sensory encoding itself but how much sensory information in the neural response affects the decision. However, in the 2018 paper, the confirmation bias was also found in a mathematical task in which (unless I am missing something) numbers are not sensed differently according to previous choices. I think the authors should clarify how viable is the alternative explanation that is falsified by this study.

Thank you for raising this important point. Your comment made us realize that the motivation of our current study did not become sufficiently clear in the previous Introduction. To clarify this point, we have now revised the Introduction and also included formal model simulations of alternative mechanisms that solidify the conceptual intuitions (Figure S5).

In what follows, we elaborate on the rationale, taking more space than in the manuscript to unpack it in full. Neural data analyzed precisely in the way we did here (dissociating neural encoding and readout) are, in fact, critical for disambiguating between two scenarios, both of which are consistent with the behavioral results reported in our previous paper that you are referring to (Talluri et al., 2018). This previous study showed that sensory or numerical information has more (less) weight on a final estimate when that information is consistent (inconsistent) with a previous categorical choice.

Critically, the behavioral effects reported by Talluri et al. (2018), in both the perceptual and the numbers task, could result from two distinct effects of a choice on the processing of subsequent information: (i) modulating the strength/gain of neural responses to subsequent (sensory or numerical) representations depending on consistency; (ii) altering the readout of these representation on downstream computations of the estimate. Both are top-down effects, but they target different stages in the processing chain. These two mechanisms could not be distinguished by the analyses reported in the (Talluri et al., 2018) paper. If anything, the best-fitting behavioral model in this paper (termed "Selective Gain") describes a multiplicative modulation of the weighting of sensory/numerical information, consistent with the effects of feature-based attention on enhancing neural representations – so a form of (i). See also the comment on this paper by Prat-Ortega & de la Rocha (2018), in the same issue of Current Biology, who followed the same interpretation.

The latter interpretation remained, however, speculative, and it became clear to us that a selective modulation of the readout of sensory representations could produce behavioral signatures that were indistinguishable at the level of behavior with the approaches we used in Talluri et al. (2018). This is a clear example of a situation where neural data combined with innovative analysis tools can arbitrate between distinct mechanisms underlying a behavioral phenomenon.

Our new model simulations presented in Figure S5 solidify these intuitions. In the model, stimulus samples are encoded linearly in the strength of neural activity in two neural populations, and the resulting neural activity pattern is then decoded to produce an estimate, by applying linear weights to the activity

of each population. Encoding and decoding are both corrupted by separate sources of noise. We consider three possible types of changes in neural activity with inconsistent samples. In the first case, the encoding deteriorates (noise increases), so the sample is less precisely encoded. In the second and third cases, the readout deteriorates, in two different ways: decoding noise increases, or (without any change in encoding or decoding noise), the encoding and readout weights become mismatched.

Critically, we show that each of these three cases give rise to the observed effect in the psychophysical kernels (less contribution of inconsistent samples to estimate). In other words, all three cases are very viable explanations for the psychophysical kernel effects in our current study as well as the results reported in Talluri et al. (2018) – and this includes the deterioration of encoding precision that we falsify in this study. The pattern of neural measures we observe in the brain is most consistent with the third case (mismatched readout) – and this holds both for the information theoretic measures from our original submission as well as new linear measures based on regression we have added in revision.

In sum, the scenario of a modulation of the sensory representation was a very viable alternative and, in fact, the one we promoted by Talluri et al. (2018). Recordings of brain activity, in conjunction with the analysis tools developed in the lab of Stefano Panzeri, were decisive for falsifying this alternative and instead supporting the idea that representations are used in the brain in a manner that depends on their consistency with previous choices. We hope that our textual revisions and the simulation results clarify this point in a concise manner.

R3.2. The measure on which the paper's main claims are based ('intersection information') is hard to grasp. Conceptually, it was unclear how the interpretation that it quantifies 'the amount of stimulus information encoded about the stimulus S by the neural signal R that informed the behavioral estimation report E' (lines 584-7).

Thank you for pointing us to this lack of clarity. The simulations reported in the new Supplemental Figure S5 also help to clarify this (see also description in previous reply). Furthermore, we have added a conceptual schematic to Figure 4 illustrating the (mis)match between neural codes for stimulus and estimates that produces differences in readout. Both additions combined explicitly convey the difference in readout in terms of a match (for consistent) or mismatch (for inconsistent) in the codes for stimulus samples and the behavioral estimate in the patterns of neural population activity.

As with any complex method, it would be reassuring to see the underlying data (source-reconstructed MEG activity).

We now show the time course of the source-reconstructed MEG response averaged across all vertices from dorsal visual cortex (Figure S6).

But here, one can only blindly trust that the method does its job. Methodologically, it seems that this 'intersection information' is computed with as low as 4 trials per bins and is normalized by a baseline obtained by only 20 permutations. One could suspect that this is a very unstable measure.

This comment raises two issues: (i) using only 20 permutations to construct the null hypothesis distribution, and (ii) whether the number of trials per bins used to sample the distributions are sufficient for

accurate information estimates. In what follows, we will address both specific issues in turn, followed by some general remarks on the stability of the measures we present.

Number of shuffles:

We have now recomputed the baseline correction with 100 shuffles also for intersection information (the shuffle correction for mutual information values was already before based on 100 shuffles). We have updated all figures (main and Supplementary) to which this pertains correspondingly. The main results are unaffected, with one minor exception: the across-participants correlation between consistency effects in psychophysical kernel and intersection information in Figure 5 is now only borderline-significant for Dorsal Visual Cortex and the Cue condition. At the same time, this correlation became even stronger for Inferior Parietal Cortex in the Choice condition.

We have performed this shuffle correction for all analyses that compare information measures between consistent and inconsistent conditions – i.e., Figure 2,4, and 5. For the visualization of overall information measures pooled across those conditions in Figure 3, we kept the original version based on $N=20$ shuffles. This is because for this condition-independent characterization of information over time and cortical space, we additionally subtracted yet another baseline measure: the average information values obtained during the baseline interval before the onset of the stimulus sequence. We realized that this conservative approach had not been mentioned in the former figure caption, so we have now added it. Also, we would like to highlight that the whole-brain analyses with 100 shuffles run for several weeks even on our reasonably large computer cluster.

Numbers of trials per bin used to compute empirical distributions:

First, our previous work (e.g. (Panzeri et al., 2007; Panzeri & Treves, 1996)) and the work of many other groups (e.g. (De Ruyter Van Steveninck et al., 1997; Strong et al., 1998)) have repeatedly documented that mutual information estimates based on discrete probabilities are accurate (in particular, unbiased) when there are at least 4 or 5 trials per bin. We have used information estimated with this sampling regime in many empirical papers (e.g. the recent papers (Emanuel et al., 2021; Kuan et al., 2024; Runyan et al., 2017)). We have also carefully validated the accuracy of estimates in this sampling regime for the PID redundancy measure used for intersection information (Koçillari et al., 2024).

Second, your comment made us realize that our description was not sufficiently clear, and partially misleading, about this point in the previous version of the manuscript. Indeed, the binning with 3 bins was chosen as it gave amply sufficient observations per bin for all conditions and participants, yet enough bins to obtain conservative and meaningful numbers. We have now clarified the description as follows (p. 22, from line 721):

“For the less sampled analyses (i.e., those focusing on the “near-zero trials”, splitting by sample consistency, and pooling across two sample positions (e.g., 7 and 8), the median trial numbers per computed information measure were 207 (range: 84 - 263) for the Cue condition and 191 (range: 87 - 257) for the Choice condition, respectively. These trial counts yielded at least 9 observations per bin (median 22 observations) for all information measures reported (for both Cue and Choice conditions). For the computations of neural information measures across all trials (Figure 3), the median number of observations per bin was 31, for both Cue and Choice.”

So, indeed, each bin contributing to all measures contributing to our reported results was populated by at least 9 observations and substantially more in the vast majority of cases. We thank you for highlighting this and apologize for any confusion our text may have caused.

Third, we further consolidated our results in two ways. 1) We simulated the information calculation of a tri-variate Gaussian process (sample, estimate, neural activity) as a function of the number of trials. We found that, when using 3 bins to discretize each quantity and using the shuffle-subtraction limited-sampling bias correction procedure employed in this paper, the ground truth value of the information measure was obtained even from very limited total number of trials, and that having 100 trials (half of those on our empirical data) or more was amply sufficient for information estimates. 2) We also demonstrated in the empirical data that the overall results are stable when computed for different number of bins. These simulation results and control analyses are now included in Figure S12 and reported in Methods (p. 22, from line 722):

“We tested the stability and validity of the information measures presented in this paper in two ways. First, we verified empirically that the qualitative patterns (time courses and cortical distributions) of all information theoretic measures assessed here were stable over a range of bin numbers (3, 5, 7, 9 bins) for both task conditions (Figure S12a-f, compare with Figure 3). Second, we performed simulations, in which we computed the information measured as function of the number of simulated trials and showed that the amount of data used in our study was comfortably sufficient to achieve a correct value of all the information quantities with the limited-sampling bias corrections used in this paper (Figure S12g-i).”

General remarks:

Please also note that, to be further conservative with sampling, we also removed from the most under-sampled analyses one participant who had less than 81 total trials for the inconsistent sample condition. Although this does not affect the results, we feel confident that each calculation presented in this paper is reasonably accurate.

Finally, we would like to highlight that the primary purpose of Figure 3 is to provide an empirical demonstration that our measures are plausible in before diving into the analyses of the neural effects of interest (consistency). Indeed, this is why we chose to show the overall stimulus information and intersection information in cortical activity in Figure 3. This figure clearly demonstrates that all information measures exhibit meaningful cortical distributions (i.e., peaking in early visual cortex and strong in all retinotopically organized visual cortical areas; Figure 3b) and well-behaved time courses in our a priori defined region of interest (Figure 3c), as well as all other visual cortical field maps (data not shown, to avoid clutter). The same holds for intersection information (Figure 3f,g).

R3.3. Neural measures are also computed in a task with fast presentations of stimuli (1 sample every 150 ms?) which I assume induces overlapping neural responses.

Your assumption is correct: the neural stimulus responses within each of the two intervals are overlapping (not however across intervals, see next point). This is evident in the evoked responses shown in

the new supplemental figure referred to above (Figure S6). However, due to the substantial sample-to-sample variability of the stimuli, we can still track neural sample information individually for each sample position (Figure 3c). This matches our results from previous decoding/encoding approaches to neural stimulus responses in similar sequential presentation designs (Murphy et al., 2021; Wilming et al., 2020).

In Figure 3, it seems that the effect lasts between about 0.1 to 0.5s post-stimulus onset and therefore the measures are also affected by the pre-choice activity generated by samples 5 and 6 that are not necessarily independent from samples 7 and 8 because of the subsampling of consistent and inconsistent samples, right? Considering the complexity of the measures I could not convince myself that the data support the conclusions. This concern applies much less to the analyses done on behavior only, for which the authors used additional simulations (lines 608 - 634) to test for some possible confounds. A similar analysis with the neural data would have helped convince.

Our task design entailed a fixed 1.75 s interval between the first interval (i.e., after sample 6) and the second interval (i.e., before sample 7). During this fixed interval, participants saw the cue (Cue condition) or made their binary choice (Choice condition). This delay of 1.75 s was sufficient for any information about sample 6 in the neural responses, as well as the corresponding intersection information, to decay back down to baseline levels. In Figure R2, we plot the time course of $II(S;R;E)$ for Dorsal Visual Cortex across the complete trial, showing this decay.

Figure R2. Time course of intersection information $II(S;R;E)$ in dorsal visual cortex across a complete trial. Dashed red vertical lines indicate the start of each sample. (a) Cue condition. (b) Choice condition. Black vertical line in (b) indicates the median response time for the binary choice, and gray box in (a) indicates the interval of the cue presentation. The intersection information $II(S;R;E)$ for samples at different positions in the sequences is coded in different grayscale. $II(S;R;E)$ for all samples from first interval including sample 6 has decayed back down to baseline by the arrival of the sample 7.

For completeness, we now also include a new Supplemental Figure (Figure S10) showing that, correspondingly, the consistency effects on $II(S;R;E)$ are not evident for sample positions 5,6.

Minor

- Why were the conditions cue/choice not randomized and counterbalanced?

We intentionally decided for this fixed order to prevent participants from forming internal categorical judgements in the Cue condition. Thank you for raising this point – we had failed to motivate this decision explicitly in our initial submission and have now done so.

Indeed, previous experiments in our laboratory with different variants of this task had shown that, once participants practice the task with an intermittent choice, they are likely to continue forming categorical judgements even when they are no longer asked to report those in the task at hand. Such a tendency would obviously produce very similar effects for Choice and Cue conditions, but for trivial reasons. The fact that we now observe very similar patterns even in our current sequential design demonstrates that the similarity is there for deeper reasons: the brain seems to use its own categorical choices for further evidence processing in a manner that corresponds to the way it uses external categorical cues.

This is now motivated in Methods.

- Line 600: is it not rather the '*estimation* E' than the 'stimulus E'? This paragraph is especially hard to follow.

Thank you for spotting this. This was indeed a typo. Fixed.

References

- Britten, K. H., Newsome, W. T., Shadlen, M. N., Celebrini, S., & Movshon, J. A. (1996). A relationship between behavioral choice and the visual responses of neurons in macaque MT. *Visual Neuroscience*, *13*(1), 87–100. <https://doi.org/10.1017/S095252380000715X>
- Cheadle, S., Wyart, V., Tsetsos, K., Myers, N., de Gardelle, V., Hecce Castañón, S., & Summerfield, C. (2014). Adaptive Gain Control during Human Perceptual Choice. *Neuron*, *81*(6), Article 6. <https://doi.org/10.1016/j.neuron.2014.01.020>
- Chicharro, D., Panzeri, S., & Haefner, R. M. (2021). Stimulus-dependent relationships between behavioral choice and sensory neural responses. *eLife*, *10*, e54858. <https://doi.org/10.7554/eLife.54858>
- De Ruyter Van Steveninck, R. R., Lewen, G. D., Strong, S. P., Koberle, R., & Bialek, W. (1997). Reproducibility and Variability in Neural Spike Trains. *Science*, *275*(5307), 1805–1808. <https://doi.org/10.1126/science.275.5307.1805>
- Emanuel, A. J., Lehnert, B. P., Panzeri, S., Harvey, C. D., & Ginty, D. D. (2021). Cortical responses to touch reflect subcortical integration of LTMR signals. *Nature*, *600*(7890), 680–685. <https://doi.org/10.1038/s41586-021-04094-x>
- Kiani, R., Cueva, C. J., Reppas, J. B., & Newsome, W. T. (2014). Dynamics of Neural Population Responses in Prefrontal Cortex Indicate Changes of Mind on Single Trials. *Current Biology*, *24*(13), 1542–1547. <https://doi.org/10.1016/j.cub.2014.05.049>
- Kiani, R., Hanks, T. D., & Shadlen, M. N. (2008). Bounded Integration in Parietal Cortex Underlies Decisions Even When Viewing Duration Is Dictated by the Environment. *The Journal of Neuroscience*, *28*(12), 3017–3029. <https://doi.org/10.1523/JNEUROSCI.4761-07.2008>
- Koçillari, L., Lorenz, G. M., Engel, N. M., Celotto, M., Curreli, S., Blanco Malerba, S., Engel, A. K., Fellin, T., & Panzeri, S. (2024). *Sampling bias corrections for accurate neural measures of redundant, unique, and synergistic information*. <https://doi.org/10.1101/2024.06.04.597303>
- Kuan, A. T., Bondanelli, G., Driscoll, L. N., Han, J., Kim, M., Hildebrand, D. G. C., Graham, B. J., Wilson, D. E., Thomas, L. A., Panzeri, S., Harvey, C. D., & Lee, W.-C. A. (2024). Synaptic wiring motifs in posterior parietal cortex support decision-making. *Nature*. <https://doi.org/10.1038/s41586-024-07088-7>
- Law, C.-T., & Gold, J. I. (2008). Neural correlates of perceptual learning in a sensory-motor, but not a sensory, cortical area. *Nature Neuroscience*, *11*(4), 505–513. <https://doi.org/10.1038/nn2070>
- Leopold, D. A. (2003). Very Slow Activity Fluctuations in Monkey Visual Cortex: Implications for Functional Brain Imaging. *Cerebral Cortex*, *13*(4), 422–433. <https://doi.org/10.1093/cercor/13.4.422>
- Mante, V., Sussillo, D., Shenoy, K. V., & Newsome, W. T. (2013). Context-dependent computation by recurrent dynamics in prefrontal cortex. *Nature*, *503*(7474), 78–84. <https://doi.org/10.1038/nature12742>
- Murphy, P. R., Wilming, N., Hernandez-Bocanegra, D. C., Prat-Ortega, G., & Donner, T. H. (2021). Adaptive circuit dynamics across human cortex during evidence accumulation in changing environments. *Nature Neuroscience*, *24*(7), 987–997. <https://doi.org/10.1038/s41593-021-00839-z>
- Neri, P., & Heeger, D. J. (2002). Spatiotemporal mechanisms for detecting and identifying image features in human vision. *Nature Neuroscience*, *5*(8), 812–816. <https://doi.org/10.1038/nn886>

- Panzeri, S., Senatore, R., Montemurro, M. A., & Petersen, R. S. (2007). Correcting for the Sampling Bias Problem in Spike Train Information Measures. *Journal of Neurophysiology*, *98*(3), 1064–1072. <https://doi.org/10.1152/jn.00559.2007>
- Panzeri, S., & Treves, A. (1996). Analytical estimates of limited sampling biases in different information measures. *Network: Computation in Neural Systems*, *7*(1), 87–107. <https://doi.org/10.1080/0954898X.1996.11978656>
- Peixoto, D., Verhein, J. R., Kiani, R., Kao, J. C., Nuyujukian, P., Chandrasekaran, C., Brown, J., Fong, S., Ryu, S. I., Shenoy, K. V., & Newsome, W. T. (2021). Decoding and perturbing decision states in real time. *Nature*, *591*(7851), 604–609. <https://doi.org/10.1038/s41586-020-03181-9>
- Pica, G., Piasini, E., Safaai, H., Runyan, C., Harvey, C., Diamond, M., Kayser, C., Fellin, T., & Panzeri, S. (2017). Quantifying how much sensory information in a neural code is relevant for behavior. In I. Guyon, U. V. Luxburg, S. Bengio, H. Wallach, R. Fergus, S. Vishwanathan, & R. Garnett (Eds.), *Advances in Neural Information Processing Systems* (Vol. 30, pp. 3686–3696). Curran Associates, Inc. <https://proceedings.neurips.cc/paper/2017/file/a9813e9550fee3110373c21fa012eee7-Paper.pdf>
- Prat-Ortega, G., & Rocha, J. D. L. (2018). Selective Attention: A Plausible Mechanism Underlying Confirmation Bias. *Current Biology*, *28*(19), R1151–R1154. <https://doi.org/10.1016/j.cub.2018.08.024>
- Runyan, C. A., Piasini, E., Panzeri, S., & Harvey, C. D. (2017). Distinct timescales of population coding across cortex. *Nature*, *548*(7665), 92–96. <https://doi.org/10.1038/nature23020>
- Strong, S. P., Koberle, R., de Ruyter van Steveninck, R. R., & Bialek, W. (1998). Entropy and Information in Neural Spike Trains. *Physical Review Letters*, *80*(1), 197–200. <https://doi.org/10.1103/PhysRevLett.80.197>
- Talluri, B. C., Urai, A. E., Tsetsos, K., Usher, M., & Donner, T. H. (2018). Confirmation Bias through Selective Overweighting of Choice-Consistent Evidence. *Current Biology*, *28*(19), Article 19. <https://doi.org/10.1016/j.cub.2018.07.052>
- Wilmington, N., Murphy, P. R., Meyniel, F., & Donner, T. H. (2020). Large-scale dynamics of perceptual decision information across human cortex. *Nature Communications*, *11*(1), Article 1. <https://doi.org/10.1038/s41467-020-18826-6>

NCOMMS-24-43181-A: Response to reviewers

We again thank the reviewers for their thoughtful and constructive evaluation of our manuscript. We are delighted to learn that they all support publication of our paper.

In what follows, we provide a detailed point-by-point reply to each of the reviewers' comments (printed in *blue italics*). Some reviewer points contained multiple questions/ comments.

Reviewer #1

The authors have significantly strengthened their manuscript by refining existing, and adding new analyses, and by improving clarity of their writing. I appreciate them de-emphasizing the difference between the interim cue and choice condition, as this was indeed confusing my reading of the previous version of the manuscript. Furthermore, supporting all information-based measures with measures based on linear regressions removes any doubt about whether these information-based measures were biased. Their simulations in the new Fig. S5 also provided some intuition into their results that was lacking in the previous version of the manuscript.

Overall, I have no further comments on the manuscript.

Minor:

Figure 4 legend: "See main text for and simulations in Figure S5 for details." - first "for" should be removed.

Thank you. We have corrected that Figure legend.

Reviewer #2

The authors have appropriately addressed all my comments from the first round of review, and have provided a particularly thorough response on the relationship between information theory and regression-based approaches to analysis of the data.

Reviewer #2 (Remarks on code availability):

I reviewed the code by examining the GitHub repository online, but I did not attempt to install and run the code myself. The code includes a README file, with a functioning link to the University of Hamburg repository where the data is stored, and clear links to toolboxes on which the analysis depends (e.g. for Information Theory-based analyses). The code is clearly laid out, in that the MATLAB scripts are named so that Figure X is created by FigureX_YYY.m - this will help readers who are interested in recreating a particular analysis to straightforwardly reproduce the reported figures.

Thank you.

Reviewer #3

The authors have answered all my concerns with convincing explanations, additional data/analyses and textual revisions.

I now think it is a great paper.

Thank you and congratulations.

Thank you for the kind words!